# Time course analysis of large-scale gene expression in incised muscle using correspondence analysis

**Tetsuya Horita**[1]*, **Mohammed Hassan Gaballah**[2], **Mamiko Fukuta**[1], **Sanae Kanno**[1], **Hideaki Kato**[1], **Masataka Takamiya**[3], **Yasuhiro Aoki**[1]*

**1** Department of Forensic Medicine, Nagoya City University Graduate School of Medical Sciences, Nagoya, Aichi, Japan, **2** Egyptian Forensic Medicine Authority, Ministry of Justice, Zainab, Cairo Governorate, Egypt, **3** Department of Forensic Medicine, Iwate Medical University, Shiwa-gun, Iwate, Japan

* ttyh@med.nagoya-cu.ac.jp (TH); aokiy@med.nagoya-cu.ac.jp (YA)

**Data Availability Statement:** Microarray data were deposited in the GEO database (NCBI Accession number: Series GSE140517).

## Abstract

Studying the time course of gene expression in injured skeletal muscle would help to estimate the timing of injuries. In this study, we investigated large-scale gene expression in incision-injured mouse skeletal muscle by DNA microarray using correspondence analysis (CA). Biceps femoris muscle samples were collected 6, 12, and 24 hours after injury, and RNA was extracted and prepared for microarray analysis. On a 2-dimensional plot by CA, the genes (row score coordinate) located farther from each time series (column score coordinate) had more upregulation at particular times. Each gene was situated in 6 subdivided triangular areas according to the magnitude of the relationship of the fold change (FC) value at each time point compared to the control. In each area, genes for which the ratios of two particular FC values were close to 1 were distributed along the two border lines. There was a tendency for genes whose FC values were almost equal to be distributed near the intersection of these 6 areas. Therefore, the gene marker candidates for estimation of the timing of injuries were detectable according to the location on the CA plot. Moreover, gene sets created by a specific gene and its surrounding genes were composed of genes that showed similar or identical fluctuation patterns to the specific gene. In various analyses on these sets, significant gene ontology term and pathway activity may reflect changes in specific genes. In conclusion, analyses of gene sets based on CA plots is effective for investigation of the time-dependent fluctuation in gene expression after injury.

## Introduction

The time course of wound healing or the estimation of wound age is one of the most important research subjects in forensic pathology. A number of research projects have been performed, and the majority of them have dealt with the timing of dermal injuries [1]. Skeletal muscles are, like skin, distributed throughout the whole body, and often affected in cases of fatal and serious injuries such as stab wounds and incisions. In skin wound models, different molecular diagnostic techniques have been used to evaluate the usefulness of many markers for estimating wound age, including cytokines and chemokines. However, the aging of wounds of skeletal

**Funding:** The author(s) received no specific funding for this work.

**Competing interests:** The authors have declared that no competing interests exist.

**Abbreviations:** ANOVA, One-way analysis of variance; BP, biological process; CA, corresponding analysis; CC, cellular component; Cm, Compromised; Dt, Detected; FC, fold change; GO, Gene ontology; HSD, Honestly Significant Difference; KEGG, Kyoto Encyclopedia of Genes and Genomes; MF, molecular function; NDt, Not Detected; PAGE, parametric analysis of gene set enrichment; PCA, principal component analysis.

muscle caused by sharp force injuries has not been well studied [2]. In previous studies, we devised a mouse skeletal muscle incision injury model and investigated the time-dependent dynamics of some cytokines that were selected based on the results of DNA microarray analysis of specimens 12 h post-injury [2, 3]. RNA and protein expressions of the specific cytokines during 6 to 48 h after injury were examined using quantitative reverse transcription PCR (qRT-PCR) and bead-based immunoassay, and some of those molecules were considered as possible markers for estimating wound timing [2, 3]. However, microarray analysis showed that an enormous number of biochemical markers were expressed during the early phase of wound recovery. The time course of the vast majority of these genes and the interactions between them remain unclear.

Correspondence analysis (CA) of gene expression has been employed in several microarray data analyses. Fellenberg et al. [4] obtained microarray data from Spellman et al. [5] in which they arrested the cell cycle using several experimental methods and analyzed the relationship between gene expression and each method by CA. The results showed that CA could reveal both relationships among either genes or hybridizations and between genes and hybridizations. In their insulin administration study on diabetic patients and normal controls, Tan et al. [6] found that CA of microarray data from Hansen et al. [7] could successfully divide each time point score into components dependent and independent on the disease status. The purpose of CA is to convert into a simpler data matrix without losing important information from the original data, to clarify the structure of a complex data matrix, and further to present the processing result visually [8]. In microarray research, CA can summarize data of each gene (rows) of each sample (columns) of originally high-dimensional data matrices in a low-dimensional projection as well as principal component analysis (PCA) [9]. CA forms a biplot in which rows and columns are simultaneously projected to subspaces of two or more dimensions, which reveals the association between them.

In this study, we obtained microarray data of incision injury samples of mouse skeletal muscle at 3 different time points post-injury. To visualize the time course fluctuation in gene expression on a plot, and to examine large-scale data using various analytical methods, CA was carried out on microarray data that was converted to a matrix (data type is fold change (FC) values) consisting of data of each time point (columns) by each gene (rows) as variables. Clustering a large number of genes should enable further exploration of injury time markers.

## Materials and methods

### Animal treatment for DNA microarray

Muscle samples were obtained and processed as described in a previous report [2]. Pathogen-free 8-week-old male BALB/c mice were divided into 4 groups (control, 6, 12, and 24 hours (h) after injury: n = 4 each). After nasal anesthesia of mice with isoflurane, the skin on the dorsal side of the left hind limb was shaved, and about a 1-cm incision was made on the skin using sterile straight stainless-steel scissors. Subsequently, a 5-mm incision was made in the biceps femoris muscle, and then the skin incision was closed using a silk suture. After surgery, the animals were allowed free access to food and water. At 6, 12, and 24 h after injury, mice were euthanized with a high concentration of carbon dioxide gas, and then a 3-mm thick sample of injured muscle tissue (about 30 mg) with the injury in the center was excised. As a control sample, biceps femoris muscle was collected from an uninjured mouse that was euthanized without making the injury. The animal experiment was approved by the Nagoya City University (NCU) animal ethics committee (authorization numbers: H25M-22 and H26-M02), and conducted according to the principles of laboratory animal care, and the guidelines for animal experimentation, NCU [10, 11].

## RNA extraction and DNA microarray

RNA was extracted as described in a previous report [2]. The samples were homogenized using a Taitec bead crusher (TAITEC Co., Saitama, Japan) at 2,500 rpm for over 30 sec. Samples of 6 and 24 h were outsourced to perform the microarray analysis (Oncomics Co. Ltd., Nagoya, Japan). Data of the microarray analysis of the control and 12 h groups had been collected in a previous study [2]. The RNA samples subjected to the present microarray hybridization had a concentration in the range of 29.78 to 281.45 ng/μL. The microarray was scanned using a DNA microarray Scanner (G2505C; Agilent Technologies, Santa Clara, CA).

## Normalization and quality control

Microarray data were deposited in the GEO database (NCBI Accession number: Series GSE140517). The signal value of each gene was normalized in the following four steps (TOHOKU CHEMICAL Co., Aomori, Japan). 1) When the signal value was lower than background (negative value), it was adjusted to 1, which meant that gene did not express. 2) The geometric mean of the values of 4 samples at each time was calculated as the representative value of the gene. 3) The signal values were converted to a base 2 logarithm. 4) In order to correct experimental errors between microarrays, the value of the 75th percentile of all gene signals was subtracted from that of each gene with respect to each time point under the assumption that the expression levels of most genes did not fluctuate. Quality control of microarray signal data was performed using settings recommended by Agilent Technologies. In addition, based on the flag information output from Feature Extraction Software v11 (Agilent Technologies), the features were evaluated with five flags, namely "saturated", "uniform", "positive and significant", "well above background", and "population outlier". The results were interpreted as "Not Detected (NDt)" when the flag was "not positive and significant" or "not above background", and as "Compromised (Cm)" when it was "saturated", "not uniform", or "population outlier". All other flags were considered to be compatible with "Detected (Dt)". The feature of a gene was determined as "Dt" only when flags of all 16 arrays (4 samples × 4 time points) were "Dt", and as "NDt" when at least one "NDt" was included in those of all arrays. Also, if at least one "Cm" flag was present, the feature was determined as "Cm". Genes containing only the features determined as "Dt" and "NDt" were used for analysis. The FC of gene expression was calculated using normalized non-logarithmic signal values of each gene of the control and each time point [12].

## Detection of upregulated or downregulated genes

One-way analysis of variance (ANOVA) was employed to extract genes of which expression levels significantly fluctuated between any of the time points. Subsequently, multiple comparisons were performed using Tukey's Honestly Significant Difference (HSD) test as a *post hoc* test for genes considered to be significant with ANOVA to detect genes with expression levels significantly different between the control and each time point. Furthermore, among the genes that were significant with Tukey's HSD test, gene sets were prepared by selecting genes whose FC values were upregulated or downregulated by more than 3- or 5-fold compared to the control, respectively. Gene sets that were up- or down-regulated more than 3-fold also included genes whose FC was more than 5 and less than 0.2, respectively.

## Gene ontology and pathway analysis

Gene ontology (GO) and pathway analyses were performed to indicate the biological function of gene sets in which expression fluctuated more than 3- or 5-fold compared to that of the

control. Fisher's exact test (one-sided test) was employed to examine whether the extracted gene sets contained significant numbers of genes prepared from known information (GO terms). Also, the similarity between the extracted gene sets and the gene lists classified according to Kyoto Encyclopedia of Genes and Genomes (KEGG) pathway information was investigated in the same manner. Furthermore, differences between the average fluctuation of expression of all genes included in each of the KEGG pathways and that of all genes that passed quality control were statistically examined with parametric analysis of gene set enrichment (PAGE) [13].

### Corresponding analysis and distance calculation

The intensity ratio between the control and each time was calculated using the signal values of the genes that passed quality control (TOHOKU CHEMICAL). An arctangent function was applied to the reciprocal of this ratio [9], which was converted to the radian value in a range of 0 to $\pi/2$. Compared to the conventional logarithmic transformation, this conversion method reduces the variance when the intensity ratio is $> 10$ or $< 0.1$. The radian value was further converted to degrees (0˚ to 90˚). A matrix (data in degrees) of which variables consisted of 3 time series by each gene was prepared, and a biplot was created with scores of two principal components of each gene and time series obtained by CA as two-dimensional coordinates ($x$, $y$). Furthermore, two kinds of (Euclidean) distances from each coordinate of the biplot were calculated: Distance 1: The distance between each gene and each time series score; Distance 2: The distance between each gene and the top 5 query genes whose expression was upregulated or downregulated the most at each time point (Table 1). Several sets of top 100, 300, and 1,000 genes close in distance to the principal components score of each time series or query gene were arranged under each setting, and GO and pathway analyses were performed as described above.

In all statistical tests, the obtained results were subjected to multiple test correction according to the Benjamini-Hochberg method, and differences were considered to be significant when the corrected $p < 0.05$ [12].

## Results

### DNA microarray analysis

As a result of the quality control, microarray data of a total of 55,527 genes were available for further analyses. At 6 h post injury, the expressions of 7,212 genes were significantly upregulated compared with that of the control (0 h), as were 5,361 at12 h, and 6,675 at 24 h. The numbers of significantly downregulated genes were 11,746 at 6 h, 18,956 at12 h, and 12,421 at 24 h. The number of genes showing each fluctuation pattern among 27 categories according to the fluctuation direction (up- or downregulated, or unchanged) at 3 time points is listed in Table 2. The most common pattern was that of insignificant fluctuation throughout the time, which included 22,872 genes or about 40% of all genes. As for genes with more than 3- or 5-fold change, however, most of them were upregulated. The number of genes with more than 3- or 5-fold downregulation was small until 12 h post-injury, then increased at 24 h (Table 3).

### Gene ontology analysis and pathway and gene set analysis

The most expressed category in the upregulated gene sets was GO terms in "biological process (BP)" followed by "cellular component (CC)" and "molecular function (MF)" (Table 4, S1 Table). The downregulated sets had smaller number of significant GO terms than the upregulated sets, although they were increased in number at 24 h. We mainly focused on the GO

**Table 1. Top 5 query genes whose expression were up- or downregulated at each time point, and their fold change (FC) values and coordinates on CA plot.**

| | FC at | | | Factor 1 | Factor 2 | Area in Fig 6[*] | Symbol in Fig 7[**] |
|---|---|---|---|---|---|---|---|
| | **6 h** | **12 h** | **24 h** | | | | |
| Upregulated genes | | | | | | | |
| 6 h | | | | | | | |
| Cxcl5 | **3163.200** | 2260.396 | 227.743 | 0.233 | -1.060 | D | A |
| Gm5483 | **3159.584** | 1841.431 | 115.447 | 0.258 | -1.178 | D | B |
| Ccl4 | **2871.902** | 1409.222 | 115.613 | 0.234 | -1.143 | D | C |
| Il-1β | **2140.053** | 1207.213 | 73.427 | 0.259 | -1.185 | D | D |
| S100a8 | **1399.826** | 1312.942 | 412.954 | 0.150 | -0.577 | D | E |
| 12 h | | | | | | | |
| Cxcl5 | 3163.200 | **2260.396** | 227.743 | 0.233 | -1.060 | D | A |
| Gm5483 | 3159.584 | **1841.431** | 115.447 | 0.258 | -1.178 | D | B |
| Ccl4 | 2871.902 | **1409.222** | 115.613 | 0.234 | -1.143 | D | C |
| S100a8 | 1399.826 | **1312.942** | 412.954 | 0.150 | -0.577 | D | E |
| Clec4d | 1190.241 | **1237.906** | 224.221 | 0.217 | -0.798 | C | F |
| 24 h | | | | | | | |
| Slpi | 570.131 | 968.934 | **806.127** | 0.247 | 0.126 | B | G |
| Saa3 | 274.366 | 479.682 | **464.848** | 0.254 | 0.204 | B | H |
| S100a8 | 1399.826 | 1312.942 | **412.954** | 0.150 | -0.577 | D | E |
| Cd300lf | 1332.905 | 1026.038 | **265.342** | 0.143 | -0.720 | D | I |
| Cxcl5 | 3163.200 | 2260.396 | **227.743** | 0.233 | -1.060 | D | A |
| Downregulated genes | | | | | | | |
| 6 h | | | | | | | |
| Hs3st5 | **0.029** | 0.064 | 0.084 | 0.049 | 0.036 | A | A |
| Ddit4l | **0.071** | 0.049 | 0.089 | 0.035 | 0.030 | F | B |
| Efnb3 | **0.084** | 0.317 | 0.224 | 0.106 | 0.046 | B | C |
| Lzts2 | **0.088** | 0.046 | 0.078 | 0.031 | 0.023 | E | D |
| Slc26a10 | **0.088** | 0.234 | 0.273 | 0.077 | 0.068 | A | E |
| 12 h | | | | | | | |
| Fam83d | 0.115 | **0.043** | 0.095 | 0.022 | 0.023 | E | F |
| Lzts2 | 0.088 | **0.046** | 0.078 | 0.031 | 0.023 | E | D |
| Ddit4l | 0.071 | **0.049** | 0.089 | 0.035 | 0.030 | F | B |
| Myh7 | 1.201 | **0.051** | 0.297 | -0.226 | -0.114 | E | G |
| Tet1 | 0.116 | **0.057** | 0.190 | 0.020 | 0.051 | F | H |
| 24 h | | | | | | | |
| Plcd4 | 0.248 | 0.165 | **0.031** | 0.032 | -0.039 | D | I |
| Ostn | 0.269 | 0.262 | **0.032** | 0.056 | -0.053 | D | J |
| Mettl11b | 0.206 | 0.162 | **0.039** | 0.040 | -0.026 | D | K |
| Gm6288 | 0.389 | 0.100 | **0.040** | -0.021 | -0.060 | D | L |
| Tmem233 | 0.225 | 0.190 | **0.041** | 0.044 | -0.033 | D | M |

[*]Area to which each gene belongs in Fig 6

[**]Symbols indicated in Fig 7b (for upregulated genes) or Fig 7c (downregulated genes)

terms that were related to the processes of inflammation and wound healing involving myoblasts because we assumed that making the incision would initiate such processes. The "BP" category in the upregulated set mainly showed GO terms associated with inflammatory response, cytokines, and wound healing (Table 5). The GO terms associated with skeletal

**Table 2. The number of genes showing each fluctuation pattern.**

| 6, 12, 24 h | Number | 6, 12, 24 h | Number | 6, 12, 24 h | Number |
|---|---|---|---|---|---|
| U, U, - | 741 | U, D, D | 25 | U, -, D | 11 |
| U, -, U | 1647 | D, U, D | 60 | U, D, - | 1068 |
| -, U, U | 830 | D, D, U | 50 | -, U, D | 74 |
| U, U, D | 33 | -, -, D | 1509 | D, U, - | 113 |
| U, D, U | 565 | -, D, - | 7168 | -, D, U | 81 |
| D, U, U | 61 | D, -, - | 1682 | D, -, U | 147 |
| U, -, - | 675 | -, D, D | 2186 | U, U, U | 2447 |
| -, U, - | 1002 | D, -, D | 1820 | -, -, - | 22872 |
| -, -, U | 847 | D, D, - | 1110 | D, D, D | 6703 |

"U: significantly upregulated, D: significantly downregulated,

-: not fluctuated, at each time point (6, 12, 24 h post-injury in order). "

muscle and myoblasts (*e.g.*, GO: 0035914, GO: 0045445) were significant mainly among the 3- or 5-fold upregulated set at 6–12 h. For example, *Myod1*, which is a gene involved in myotube differentiation in mice and was annotated with GO terms as described above [14], was more than 5-fold upregulated at 6 and 12 h. GO terms that were significant in "CC" and "MF" categories were primarily associated with cell membranes, or receptor activities and binding of cytokines, chemokines, and immunoglobulins. Pathways with significant differences were also predominantly selected in the upregulated gene sets (Table 4), and they were mainly related to inflammatory reactions and cytokines such as mmu04060 (Cytokine-cytokine receptor interaction) (S1 Fig). Several infection pathways such as mmu05140 (Leishmaniasis) were also involved. GO terms that were significant at all time points in both more than 3 and 5-fold upregulated sets included 647 of "BP", 17 of"CC", and 32 of "MF", whereas 17 pathways were shared by both sets (Table 6). Again, most of them were GO terms associated with inflammatory response, cell membrane, and cytokine activity, and other pathways related to inflammatory response. In the downregulated sets, significant pathways were only found in the > 3-fold set at 24 h and were mainly associated with myocardial disease and the endocrine system. Many pathways associated with the inflammatory response were indicated as significant also by PAGE at all time points.

## Corresponding analysis

Row scores of 55,527 genes by column scores of 3 time series were computed by CA, and a biplot was created (Fig 1). The cumulative contribution rate reached 100% up to the second principal component (Table 7). Assuming a triangle formed by three time series coordinates (t6-24h), all genes were distributed in an overlapping, similar, and enlarged triangular area

**Table 3. The number of genes that showed more than 3- or 5-fold expression fluctuation at each time point.**

| Fold change | Regulation | 6 h set | 12 h set | 24 h set |
|---|---|---|---|---|
| 3 | Total | 3839 | 3455 | 4974 |
| | Up | 2555 | 2277 | 2240 |
| | Down | 1284 | 1178 | 2734 |
| 5 | Total | 1600 | 1549 | 1731 |
| | Up | 1386 | 1343 | 1120 |
| | Down | 214 | 206 | 611 |

**Table 4. The number of GO terms belonging to each GO category and pathways that were significant in the GO and pathway analysis of each set with more than 3- or 5-fold fluctuation.**

| Category | Fold change | Regulation | 6 h set | 12 h set | 24 h set |
|---|---|---|---|---|---|
| Biological process | 3 | Both* | 804 | 1246 | 690 |
| | | Up | 1139 | 1532 | 956 |
| | | Down | 19 | 19 | 173 |
| | 5 | Both | 972 | 1125 | 652 |
| | | Up | 1039 | 1186 | 814 |
| | | Down | 2 | 0 | 59 |
| Cellular component | 3 | Both | 20 | 52 | 54 |
| | | Up | 38 | 76 | 56 |
| | | Down | 0 | 6 | 51 |
| | 5 | Both | 24 | 35 | 30 |
| | | Up | 27 | 46 | 23 |
| | | Down | 0 | 0 | 46 |
| Molecular function | 3 | Both | 51 | 95 | 53 |
| | | Up | 75 | 107 | 51 |
| | | Down | 3 | 22 | 44 |
| | 5 | Both | 57 | 70 | 39 |
| | | Up | 61 | 83 | 57 |
| | | Down | 0 | 4 | 38 |
| Pathway | 3 | Both | 21 | 33 | 20 |
| | | Up | 25 | 43 | 31 |
| | | Down | 0 | 0 | 15 |
| | 5 | Both | 20 | 27 | 15 |
| | | Up | 26 | 30 | 19 |
| | | Down | 0 | 0 | 0 |

*The number of significant terms or pathways in the GO and pathway analysis of combined sets of the up- and downregulated genes at each time point ("Total" set in Table 3).

(tAg). Genes of which the FC value was larger than 1 at each time point were relatively evenly distributed within tAg (Fig 2a–2c). However, most genes with an FC value of 5 or more at each time point tended to gather near the vertex of tAg. On the other hand, genes of which the FC value was smaller than 1 at each time point appeared to be distributed around the particular time series scores (Fig 2d–2f). In particular, most genes with an FC value of 0.2 or lower at each time point were localized to a narrower range around the time series score than genes with an FC value of 0.2 to 1.

Genes formed fairly clear clusters on the plot according to whether each of 3 FC values was greater than 1 (S2 Fig). In other words, all genes were classified into 27 categories according to the fluctuation patterns at each time point. Genes significantly upregulated at a particular time point were distributed throughout the region of tAg, but appeared to be gathered around the other two time series scores (Fig 3a–3c). In contrast, those that were significantly downregulated at a particular time point were predominantly gathered around the same time series scores (Fig 3d–3f). There were more downregulated genes than upregulated ones at each time point, but the distribution areas were narrower in the downregulated genes. In addition, genes that were upregulated significantly at all time points were distributed all around tAg, whereas genes without fluctuation throughout time points were mainly distributed within the inscribed circle of tAg (Fig 4a and 4b). Genes that were downregulated at all time points were mainly

**Table 5. The number of genes belonging to specific GO terms in the GO analysis of each set with more than 3- or 5-fold fluctuation.**

| GO ID | GO term | 3-fold Both* | | | 3-fold Upregulated | | | 3-fold Downregulated | | | 5-fold Both | | | 5-fold Upregulated | | | 5-fold Downregulated | | |
|---|---|---|---|---|---|---|---|---|---|---|---|---|---|---|---|---|---|---|---|
| | | 6 h | 12 h | 24 h | 6 h | 12 h | 24 h | 6 h | 12 h | 24 h | 6 h | 12 h | 24 h | 6 h | 12 h | 24 h | 6 h | 12 h | 24 h |
| GO:0006935 | chemotaxis | 152** | 171 | 157 | 128 | 153 | 122 | 24 | 18 | 35 | 98 | 119 | 91 | 93 | 116 | 80 | 5 | 3 | 11 |
| GO:0043292 | contractile fiber | 29 | 34 | 79 | 20 | 20 | 10 | 9 | 14 | 69 | 9 | 12 | 26 | 1 | 11 | 5 | 1 | 1 | 21 |
| GO:0001816 | cytokine production | 163 | 197 | 177 | 149 | 181 | 142 | 14 | 16 | 35 | 113 | 134 | 91 | 110 | 129 | 82 | 3 | 5 | 9 |
| GO:0006954 | inflammatory response | 193 | 225 | 176 | 179 | 210 | 154 | 14 | 15 | 22 | 144 | 161 | 107 | 139 | 155 | 100 | 5 | 6 | 7 |
| GO:0042692 | muscle cell differentiation | 72 | 82 | 110 | 41 | 47 | 24 | 31 | 35 | 86 | 31 | 28 | 43 | 23 | 23 | 15 | 8 | 5 | 28 |
| GO:0006936 | muscle contraction | 50 | 56 | 91 | 32 | 38 | 29 | 18 | 18 | 62 | 19 | 25 | 31 | 16 | 22 | 11 | 3 | 3 | 20 |
| GO:0061061 | muscle structure development | 123 | 135 | 168 | 70 | 79 | 41 | 53 | 56 | 127 | 51 | 52 | 64 | 41 | 42 | 23 | 10 | 10 | 41 |
| GO:0045445 | myoblast differentiation | 21 | 24 | 22 | 17 | 21 | 10 | 4 | 3 | 12 | 13 | 17 | 13 | 13 | 16 | 9 | 0 | 1 | 4 |
| GO:0007520 | myoblast fusion | 11 | 17 | 12 | 10 | 15 | 8 | 1 | 2 | 4 | 8 | 10 | 7 | 8 | 10 | 6 | 0 | 0 | 1 |
| GO:0014839 | myoblast migration involved in skeletal muscle regeneration | 2 | 3 | 2 | 2 | 3 | 2 | 0 | 0 | 0 | 0 | 0 | 0 | 0 | 0 | 0 | 0 | 0 | 0 |
| GO:0044459 | plasma membrane part | 409 | 459 | 524 | 282 | 347 | 287 | 127 | 112 | 237 | 212 | 248 | 235 | 186 | 225 | 157 | 26 | 23 | 78 |
| GO:0005102 | receptor binding | 290 | 312 | 333 | 211 | 255 | 190 | 79 | 57 | 143 | 160 | 190 | 159 | 141 | 175 | 111 | 19 | 15 | 48 |
| GO:0043269 | regulation of ion transport | 107 | 121 | 184 | 76 | 82 | 66 | 31 | 39 | 118 | 55 | 62 | 85 | 47 | 54 | 43 | 8 | 8 | 42 |
| GO:0035914 | skeletal muscle cell differentiation | 26 | 24 | 19 | 16 | 16 | 7 | 10 | 8 | 12 | 12 | 14 | 6 | 11 | 14 | 3 | 1 | 0 | 3 |
| GO:0003009 | skeletal muscle contraction | 5 | 4 | 20 | 2 | 1 | 2 | 3 | 3 | 18 | 0 | 1 | 8 | 0 | 1 | 0 | 0 | 0 | 8 |
| GO:0042060 | wound healing | 79 | 96 | 92 | 68 | 83 | 64 | 11 | 13 | 28 | 48 | 59 | 45 | 46 | 55 | 38 | 2 | 4 | 7 |

*The number of genes belonging to each GO term in the GO and pathway analysis of the combined set of the up- and downregulated genes at each time point.

**Boldface letters indicate that the particular GO terms are significant at each time point.

**Table 6. Pathways that were significant throughout the time course in both sets of genes upregulated by more than 3- and 5-fold.**

| Entry | Name |
|---|---|
| mmu04662 | B cell receptor signaling pathway |
| mmu04062 | Chemokine signaling pathway |
| mmu04060 | Cytokine-cytokine receptor interaction |
| mmu04623 | Cytosolic DNA-sensing pathway |
| mmu04640 | Hematopoietic cell lineage |
| mmu04657 | Il-17 signaling pathway |
| mmu05134 | Legionellosis |
| mmu05140 | Leishmaniasis |
| mmu04650 | Natural killer cell mediated cytotoxicity |
| mmu04064 | NF-kappa B signaling pathway |
| mmu04621 | NOD-like receptor signaling pathway |
| mmu04380 | Osteoclast differentiation |
| mmu04145 | Phagosome |
| mmu05150 | Staphylococcus aureus infection |
| mmu04668 | TNF signaling pathway |
| mmu05202 | Transcriptional misregulation in cancer |
| mmu05152 | Tuberculosis |

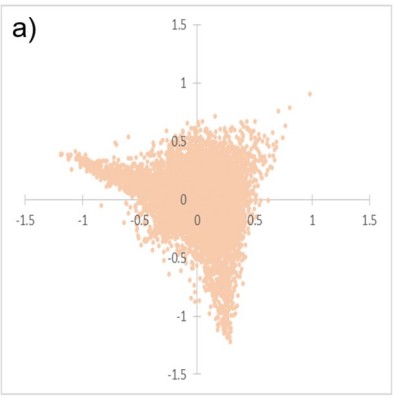
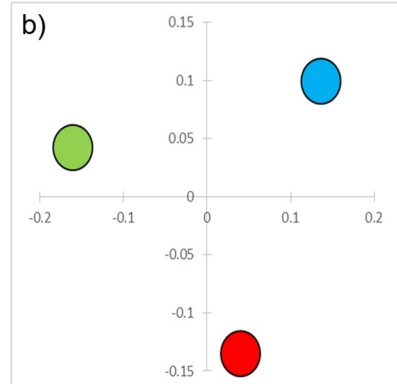

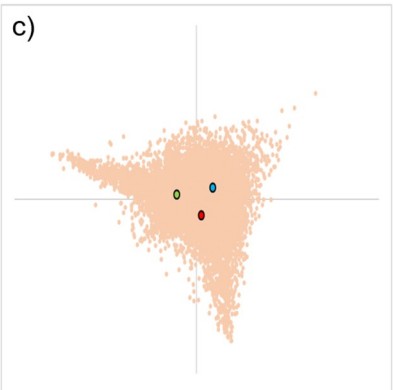

**Fig 1. Correspondence analysis (CA) plots created from microarray results.** a) The row scores of all genes are indicated by orange dots. b) The column scores of 6, 12, and 24 h are indicated by blue, green, and red dots, respectively. c) Biplot of row and column scores.

distributed in the narrower region inside t6-24h (Fig 4c). Because genes that were significantly downregulated or without fluctuation throughout time were distributed at the center of tAg, there was a tendency for many genes with large FC values at each time point to be distributed near the vertex. However, some genes even with small FC values were also located near the vertex. In addition, genes showing other fluctuation patterns were distributed in a characteristic manner. For example, there were some genes that were significantly upregulated at two time points and unchanged at the other time point, or had one time point each where they were significantly upregulated, downregulated, and without fluctuation. Among them, genes with the same fluctuation direction were plotted at symmetrical positions to each other across the center of tAg according to the combination of fluctuation behavior at each time point (Fig 4d and 4e and S3 Fig).

Based on a report showing that genes with similar expression dynamics tend to be located close to each other on a CA plot [9], the regularity of the distribution was examined in detail.

**Table 7. Eigenvalue and contribution rate in correspondence analysis.**

| Factor | Eigenvalue | Contribution | Cumulative contribution |
|---|---|---|---|
| 1 | 0.015 | 0.608 | |
| 2 | 0.010 | 0.392 | 1.000 |

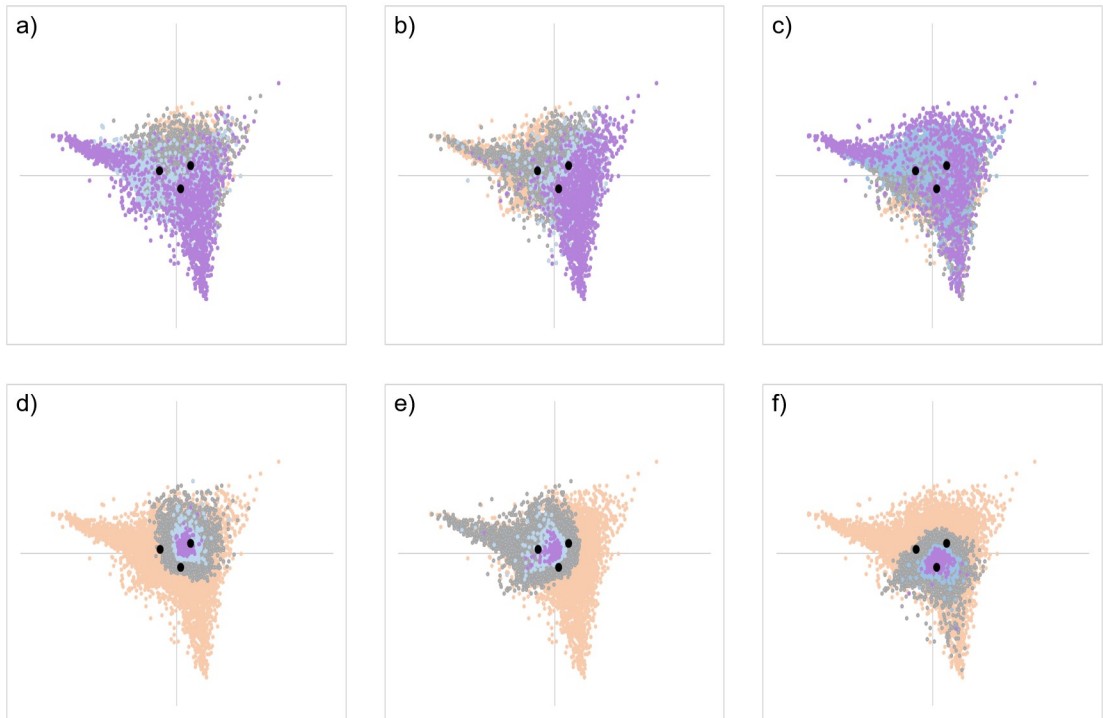

**Fig 2. CA plots of the row scores of genes that showed fluctuation at each time point.** The row scores of the CA plots are color-coded according to the magnitude relationship of fold change (FC) value of each time series. a) FC6 > 1, b) FC12 > 1, c) FC24 > 1, d) FC6 < 1, e) FC12 < 1, f) FC24 < 1, a-c). Genes with FC values of 1 < FC < 3.3 ≤ FC < 5, and FC ≥ 5 are shown in gray, blue, and purple, respectively. d-f) Genes with FC values of 0.33 < FC < 1, 0.2 < FC ≤ 0.33, and FC ≤ 0.2 are shown in gray, blue, and purple, respectively. Genes that did not satisfy any condition and the 3 time series scores are shown in orange and black dots, respectively. Assuming a triangle formed by the three time series coordinates (t6-24h), all genes were distributed in an overlapping, similar, and enlarged triangular area (tAg). Most genes with FC ≥ 5 or more at each time point tended to gather near the vertex of tAg. On the other hand, genes with FC < 1 at each time point appeared to be distributed around the particular time series scores.

On the plot, each gene was situated in 6 subdivided triangular areas according to the magnitude relationship of the FC value at each time point (Fig 5a). Each region appeared to be demarcated by 3 straight lines that intersect at one point. The genes for which the FC value at 12 h post-injury (FC12) were smallest and those of 6 h (FC6) were largest (FC12 < FC24 < FC6) were distributed in an area including the intersection coordinates (0, 0) of the first and second principal components; therefore, the 3 lines did not pass through the origin. In each area, genes for which the ratios of two adjacent FC values were close to 1 were distributed along the two border lines (S4 Fig). For example, in the region in which genes with FC6 < FC12 < FC24 were distributed, there was a tendency for genes of which FC6/FC12 was close to 1 to be located at the left end, while those for which FC12/FC24 was close to 1 were at the right end of the region (Fig 5b–5d). Moreover, genes with FC6/FC24 ≈ 1, namely FC6 ≈ FC12 ≈ FC24, were distributed near the intersection of the two straight lines forming the area of A (Fig 5e). As shown in Fig 5a, six subdivided regions of A to F were considered to be separated by three straight lines. Thus, to draw these three approximate straight lines, six genes for which the ratio of two FC values was sufficiently close to 1 were selected under the assumption that they should distribute at the end of each region (Table 8). Based on the coordinates of these genes, three approximate straight lines were created (Fig 6). The detailed relationship between each straight line and each gene was examined and is described in the S1 Appendix.

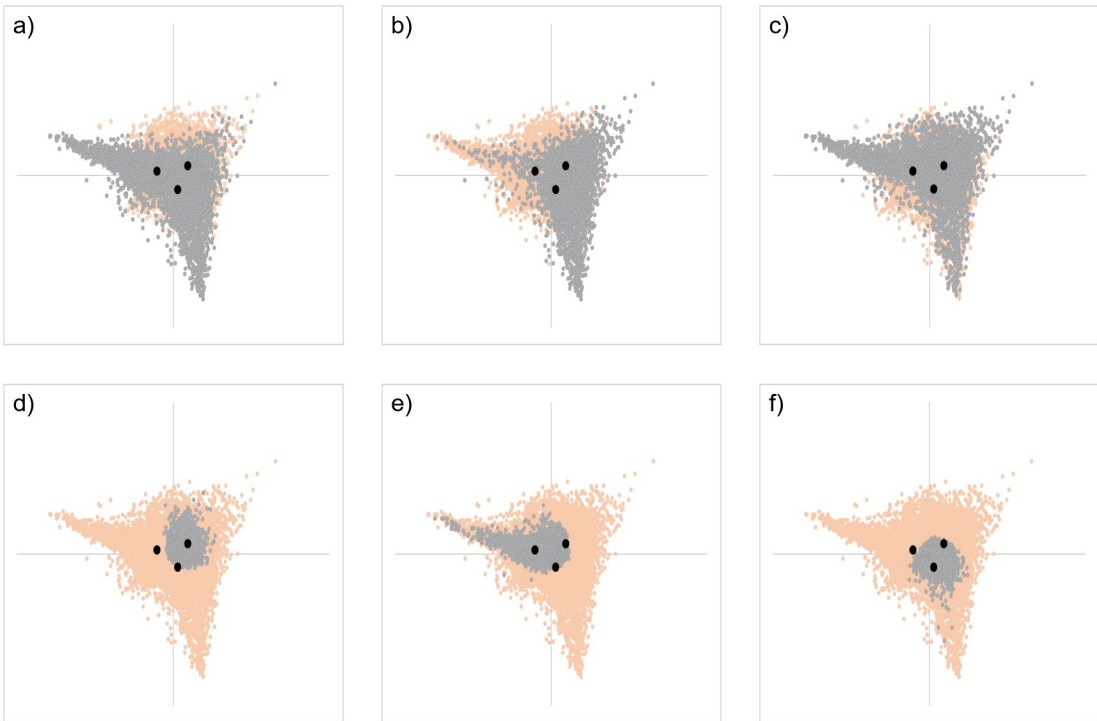

**Fig 3. CA plot of genes whose expression was significantly upregulated or downregulated at each time point.** Distribution of genes with significant upregulation at 6 h (a), 12 h (b), and 24 h (c); and downregulation at 6 h (d), 12 h (e), and 24 h post-injury is shown. Genes that did not satisfy any condition and the 3 time series scores are shown in orange and black, respectively. Genes significantly upregulated at a particular time point were distributed throughout the region of tAg but appeared to be gathered around the other two time series scores. In contrast, those that were significantly downregulated at a particular time point were predominantly gathered around the same time series scores.

### Analyses of genes close to 3 time series scores (distance 1)

In the gene sets close to the column score coordinates of 6 h, most genes were significantly downregulated or unfluctuating at 6 h (Table 9). There were no upregulated genes except for those with persistent upregulation throughout time. In the top 1,000 set, there were 186 genes whose FC6 > 1, and all of them satisfied FC12 > 1 and FC24 > 1 (Table 10). A similar tendency was observed for the genes sets close to the 12 and 24 h column score coordinates. In each set, significant GO terms and pathways were generally sparse. Among them, relatively many GO terms associated with the organelles in the "CC" category and metabolic processes in the "BP" category were found in the top 1,000 set for 6 h (Table 11). There were no significant pathways in any sets of 6 to 24 h.

### Analyses of genes close to query gene scores (distance 2)

The top 5 genes whose expression (FC value) was upregulated or downregulated the most at each time point were selected as query genes (Table 1). For the query genes at each time point, upregulated genes were mainly distributed outside t6-24h on the CA plot, whereas downregulated genes were mainly distributed inside it (Fig 7a–7c). In the query gene upregulated at 24 h, 2 genes of *Slpi* and *Saa3* and 3 genes of *S100a8*, *Cd300lf*, and *Cxcl5* were distributed apart across the horizontal axis. Although the difference in FC value between *Saa3* and *S100a8* was smaller than that between *Slpi* and *Saa3* (Table 1), the distance to *Saa3* was closer for *Slpi* than for *S100a8* on the plot. This was because the FC values of *Slpi* and *Saa3* satisfied FC6 < FC24 < FC12 (area of B of the Fig 7), while the other 3 genes satisfied FC24 < FC12 < FC6 (area of

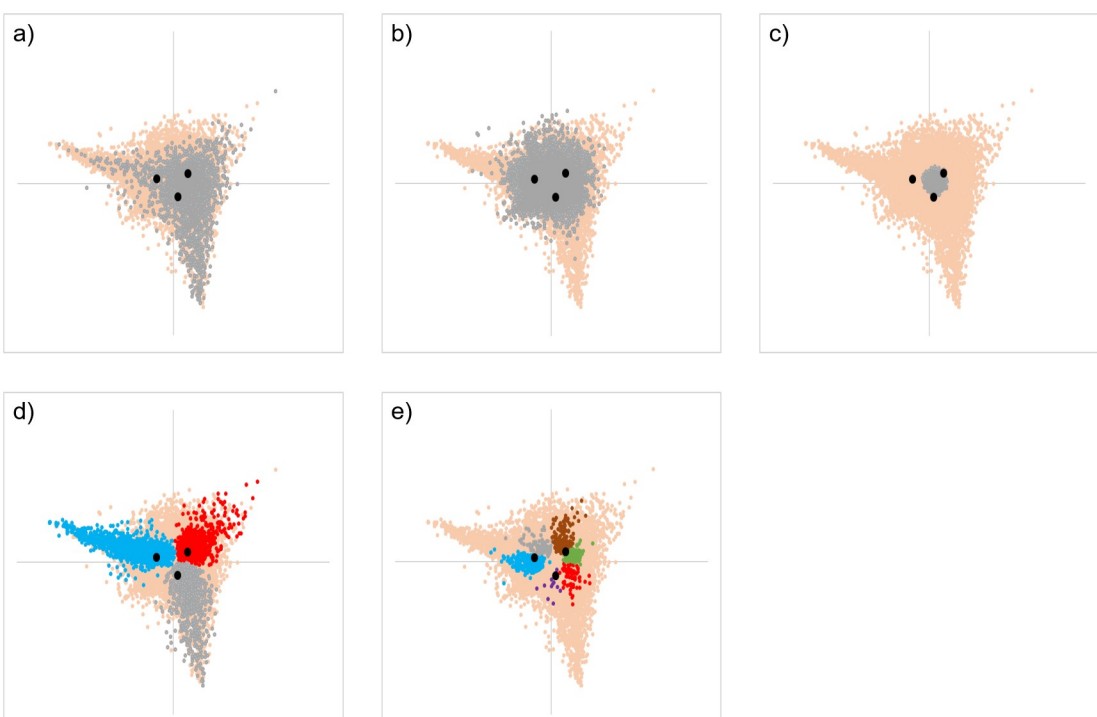

**Fig 4. Distribution of genes that showed specific time-dependent expression patterns on the CA plot (see Table 2).** a-c) Gray dots indicate the coordinates of genes with significant upregulation (a), and without fluctuation (b), and with downregulation (c) throughout the time course (presented as (U, U, U), (-, -, -), and (D, D, D) in Table 2, respectively). U, -, and D in the parenthesis are fluctuation statuses at, 6, 12, and 24 h in order. Genes that were upregulated significantly at all time points were distributed over the whole tAg region. Genes without fluctuation throughout the time course were mainly distributed within the inscribed circle of tAg. Genes that were downregulated at all time points were mainly distributed in the narrower region inside t6-24h. d) Plot of genes that have two time series with significant upregulation and one time series without fluctuation. Gray dots: (U, U, -); blue dots: (U, -, U); red dots: (-, U, U). e). Plot of genes that have one time series each that are significantly upregulated, significantly downregulated, and without fluctuation. Purple dots: (U, -, D); blue dots: (U, D, -); red dots: (-, U, D); green dots: (D, U, -); gray dots: (-, D, U); and Brown dots: (D, -, U). Genes with the same fluctuation direction were plotted at symmetrical positions with each other across the center of tAg according to the combination of fluctuation behavior at each time point. Genes that did not satisfy any condition and the 3 time series scores are shown in orange and black dots, respectively.

D of the Fig 7, Table 1). Table 12 shows the number of GO terms and pathways that were significant in all query gene sets. Similar to the more than 3 and 5-fold-upregulated sets, GO terms related to inflammatory responses and cytokines were significant in many upregulated query gene sets such as *Cxcl5*.

The significant GO terms of "BP" and "MF" series of *Slpi* and *Saa3* were fewer compared with other upregulated query genes, and those of "CC" was relatively large in number (Table 12). Among the sets of the downregulated query genes, there were smaller number of significant GO terms than those of the upregulated genes, at most 154 in the top 1,000 set of *Tmem 233*. Pathway analysis also detected those related to inflammatory responses and cytokines in many upregulated query gene sets. However, few pathways were detected with the set of *Slpi*, *Saa3*, and downregulated query genes. The details on each query genes were examined and described in the S2 Appendix.

## Discussion

In previous studies, the post-injury dynamics of 10 cytokines that were selected based on the data from a microarray analysis at 12 h post-injury were investigated with qRT-PCR from 6 to

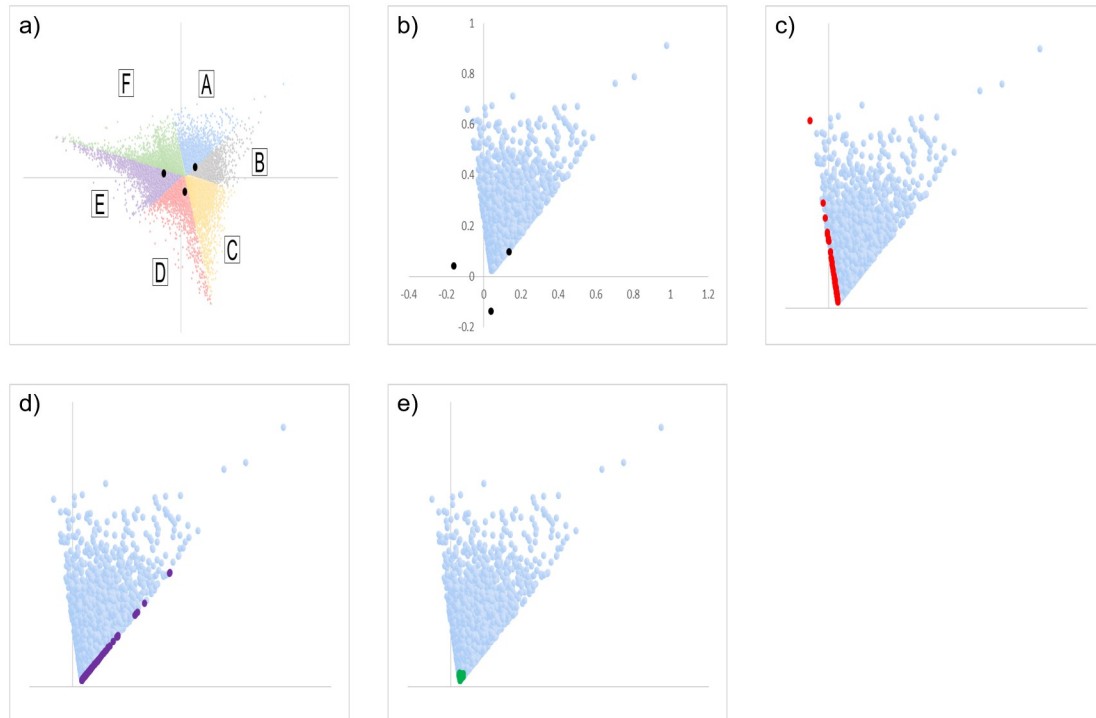

**Fig 5. Distribution of genes satisfying specific conditions in the CA plot.** a): The CA plot was color-coded according to the magnitude relationship of each FC value of the genes. Genes formed clear clusters on the CA plot according to the magnitude relationship of their FC values at each time point. Area A): FC6 < FC12 < FC24, B): FC6 < FC24 < FC12, C): FC24 < FC6 < FC12, D): FC24 < FC12 < FC6, E): FC12 < FC24 < FC6, F): FC12 < FC6 < FC24. b-e) Magnified images of area A (shown in blue dots). b) 3 column scores are superimposed by black dots. c-e) The top 100 genes for which the following values were close to 1 are highlighted c): FC12/FC6 (red), d): FC24/FC12 (purple), e): FC24/FC6 (green). Genes for which FC6/FC12 was close to 1 are located at the left end and those for which FC12/FC24 was close to 1 are at the right end of the area. The genes with FC6/FC24 ≈ 1, namely FC6 ≈ FC12 ≈ FC24, are distributed near the intersection of the two straight lines forming the area.

24 or 48 h after injury using the same mouse incised wound model [2,3]. The results of the microarray analyses of the particular genes in this study were essentially consistent with those of qRT-PCR. Thus, we believe that the microarray for each time point in this study had fairly reliable reproducibility.

The basic purpose of most microarray analyses is to elucidate biological processes or pathways that consistently show differential expression between groups of samples [15]. GO annotation and pathway information are important tools in the analysis of microarray experimental

**Table 8. Genes and their coordinates employed to draw the 3 approximate straight lines dividing the 6 areas\*.**

| Probe Name | Symbol | Factor 1 | Factor 2 | Linear equations | Line No.\* |
|---|---|---|---|---|---|
| A_55_P2054857 | Ube2v2 | 0.050 | -0.013 | y = -4.903 x 0.232 | *1* |
| A_55_P2027436 | Bri3 | 0.026 | 0.105 | | |
| A_30_P01025341 | | 0.023 | 0.005 | y = 0.923 x − 0.017 | *2* |
| A_55_P1955467 | Bbs1 | 0.101 | 0.077 | | |
| A_30_P01024768 | | 0.039 | 0.024 | y = -0.302 x + 0.035 | *3* |
| A_52_P504743 | Cdyl | 0.136 | -0.006 | | |

\*Presented in Fig 6.

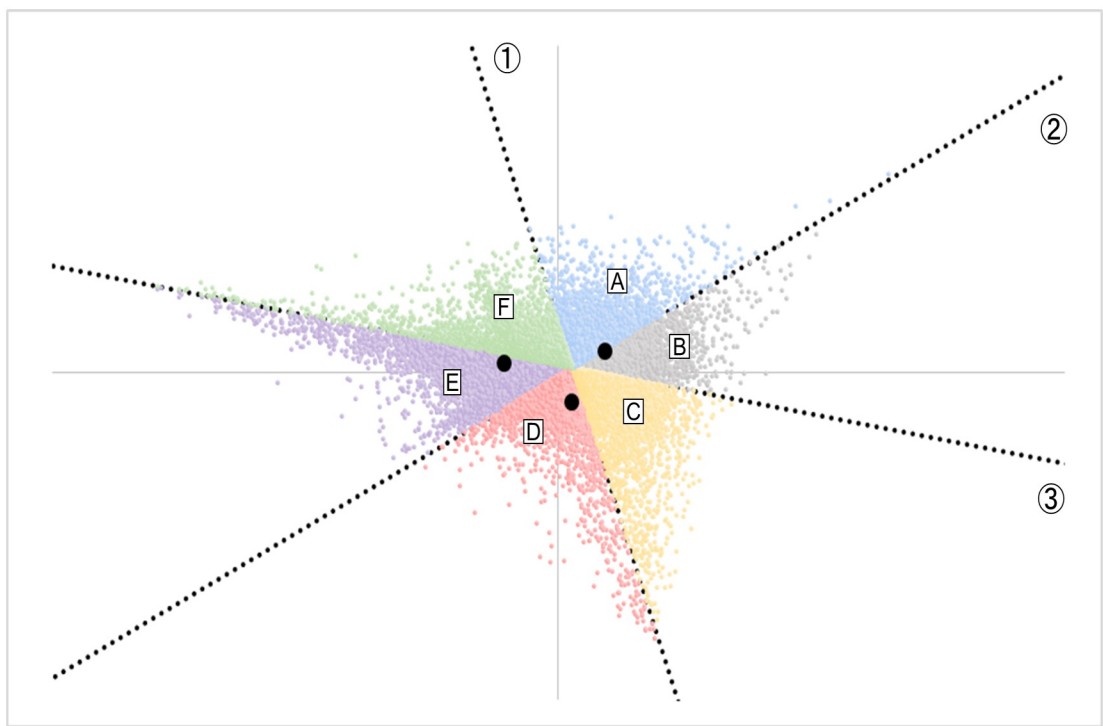

**Fig 6. Three straight lines, 1 to 3, calculated from the row scores of the specific genes (see Table 8) are drawn on the CA plot.** The lines 1 to 3 are the boundaries of the magnitude relationship between FC6 and FC12, FC12 and FC24, and FC24 and FC6, respectively. The equations for each line are as follows: 1: y = -4.903 x + 0.232; 2: y = 0.923 x − 0.017; 3: y = -0.302 x + 0.035. Black dots indicate the 3 column scores.

results [16–20]. In many cases, researchers select genes that are upregulated or downregulated more than a specific threshold in a microarray analysis, then perform GO and pathway analysis [20–26]. However, Yang et al. [27] suggested that genes fluctuating less at a significant level could also be of importance for the understanding of specific reactions. Shen et al. [20]

**Table 9. The number of genes with each fluctuation pattern in the top 100 to 1000 gene sets whose distance was close to the column score of 6 h on the CA plot.**

| 6, 12, 24 h | 100 set | 300 set | 500 set | 1000 set |
|---|---|---|---|---|
| -, U, U | 8 | 26 | 48 | 96 |
| D, U, U | 2 | 3 | 5 | 8 |
| -, U, - | 0 | 3 | 6 | 16 |
| -, -, U | 0 | 0 | 1 | 7 |
| D, -, - | 34 | 106 | 177 | 350 |
| D, -, D | 1 | 10 | 18 | 50 |
| D, D, - | 0 | 3 | 6 | 15 |
| D, U, - | 2 | 5 | 7 | 16 |
| D, -, U | 0 | 2 | 3 | 3 |
| U, U, U | 6 | 18 | 30 | 57 |
| -, -, - | 43 | 105 | 162 | 302 |
| D, D, D | 4 | 19 | 37 | 80 |

"U: significantly upregulated, D: significantly downregulated,

-: not fluctuated, at each time point (6, 12, 24 h post-injury in order). "

**Table 10. The number of genes with each magnitude relationship at each time point in the top 100 to 1000 gene sets whose distance was close to the column score of 6 h on the CA plot.**

| Fold change | 100 set | 300 set | 500 set | 1000 set |
|---|---|---|---|---|
| (FC6, FC12, FC24) < 1 | 36 | 126 | 220 | 450 |
| (FC6, FC24) < 1 < FC12 | 7 | 16 | 28 | 68 |
| (FC6, FC12) < 1 < FC24 | 0 | 4 | 9 | 30 |
| FC6 < 1 < (FC12, FC24) | 38 | 106 | 153 | 266 |
| (FC12, FC24) < 1 < FC6 | 0 | 0 | 0 | 0 |
| FC24 < 1 < (FC6, FC12) | 0 | 0 | 0 | 0 |
| FC12 < 1 < (FC6, FC24) | 0 | 0 | 0 | 0 |
| 1 < (FC6, FC12, FC24) | 19 | 48 | 90 | 186 |

performed PCA to visualize time-dependent expression pattern of microarray data of tibialis anterior muscle after peripheral nerve injury in rats. On the other hand, Yano et al. [9] employed CA for microarray analysis to estimate genes related to breast cancer or housekeeping genes by measuring the distances between each gene and artificial marker genes on CA plots, and performed GO analysis on the genes related to breast cancer. They considered that it was better to evaluate genes detected from the entire microarray dataset than to detect and evaluate candidate genes using thresholds. They showed also that up- or down-regulated genes could be predicted only with CA using the arctangent function, and that PCA was not appropriate for the clustering of genes according to their expression patterns using any index [9].

In this study, genes were successfully clustered based on the magnitude relationship of FC values of 3 time series with a CA plot, which effectively visualizes the data of time course experiments. GO and pathway analyses of the query gene set based on the CA plot revealed GO terms and pathways that many genes showing fluctuation patterns similar to the query genes belong to. Most of the upregulated genes shown in Table 1 are involved in inflammation or other immune system functions [28–44]. The large FC values of these genes in our microarray results were considered to be consistent with their bioactivity after injury (except for *Gm5483*, which is less well studied than other cytokines). In the top 100 to 1,000 gene sets created by designating *Cxcl5* as a query gene, there were two types of genes in the set that were included in gene lists of the significant GO terms and pathways, since *Cxcl5* is located close to a borderline and around the vertex of tAg as seen in Fig 6. Although the number of genes was small near the vertex of tAg, many GO terms and pathways were significant. This was due to the paucity of significantly downregulated genes. Most of the genes in the set were those with relatively large upregulation at each time point (Figs 2, 3 and 4a–4c). Therefore, almost all GO terms and pathways that were significant in these gene sets were considered to be upregulated

**Table 11. The number of GO terms belonging to each GO category that were significant in the GO analysis of the top 100 to 1000 gene sets whose distances were close to each column score on the CA plot.**

| GO terms | Column score | 100 set | 300 set | 1000 set |
|---|---|---|---|---|
| Biological process | 6 h | 0 | 45 | 48 |
| | 12 h | 0 | 0 | 0 |
| | 24 h | 0 | 0 | 0 |
| Cellular component | 6 h | 4 | 10 | 43 |
| | 12 h | 0 | 0 | 0 |
| | 24 h | 0 | 0 | 4 |
| Molecular function | 6 h | 0 | 0 | 4 |
| | 12 h | 0 | 0 | 0 |
| | 24 h | 0 | 0 | 0 |

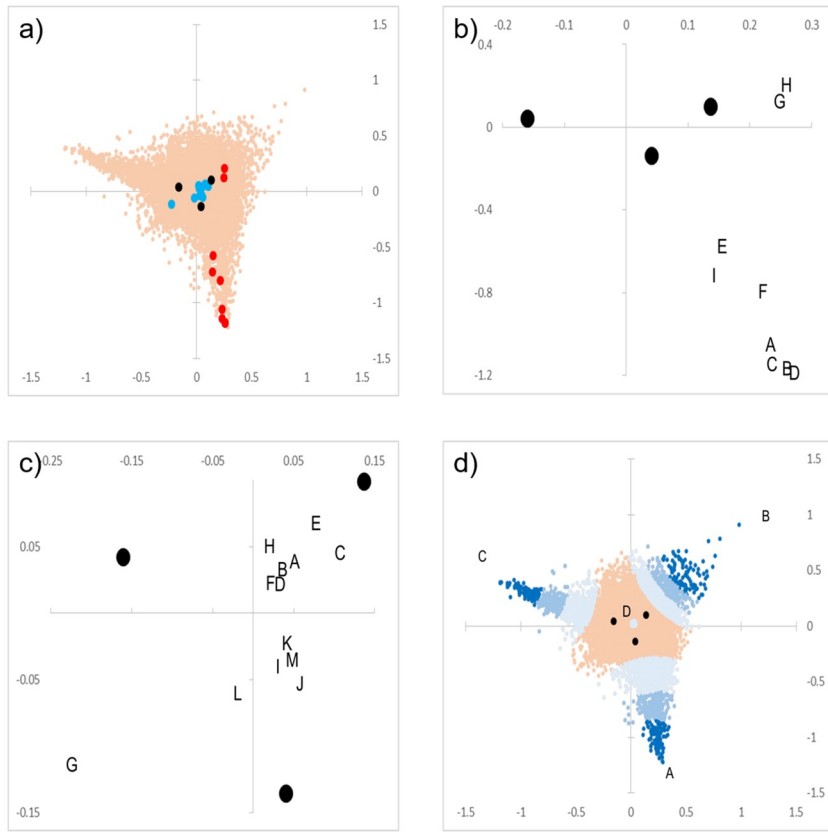

**Fig 7. Distribution on the CA plot of the query genes.** a) The query genes that were upregulated and downregulated are shown in red and blue, respectively. b) The query upregulated genes and time series scores. A: *Cxcl5*, B: *Gm5483*, C: *Ccl4*, D: *Il-1β*, E: *S100a8*, F: *Clec4d*, G: *Slpi*, H: *Saa3*, I: *Cd300lf*. Upregulated genes are mainly distributed outside t6-24h on the CA plot. c): The query downregulated gene and time series scores. A: *Hs3st5*, B: *Ddit4l*, C: *Efnb3*, D: *Lzts2*, E: *Slc26a10*, F: *Fam83d*, G: *Myh7*, H: *Tet1*, I: *Plcd4*, J: *Ostn*, K: *Mettl11b*, L: *Gm6288*, M: *Tmem233*. Downregulated genes are mainly distributed inside t6-24h on the CA plot. d): Plot of top 100–1,000 genes close to the 4 query genes. A: *Cxcl5*, B: *Arg1*, C: *Ly6f*, D: *Fam83d*. The top 100 genes close to each query genes are indicated by dark blue dots, and 300 genes are by blue and 1,000 genes by light blue. The distribution of the top 100–1,000 closest to *Fam83d* is spread to a very small area, unlike the other 3 genes. In a) and d), genes that did not satisfy the conditions are shown in orange dots.

at each time point. However, there were few significant GO terms and pathways in the *Ly6f* sets that were mapped near the vertex of tAg (Fig 7d and Table 12), suggesting that there were few GO terms and pathways to which the gene with the highest expression at 12 h belonged. On the other hand, the top 300 set of *Slpi* around which more genes were located than the vertex areas was composed of genes of only a single area (area B (FC6 < FC24 < FC12) shown in Fig 6). Even in the center of tAg, unless a gene located very close to point P is selected as the query gene, the gene set will not include more than two areas in Fig 6. In the central part of tAg, we assumed that many genes had few or no GO term annotations, or that these genes had various and divergent GO term annotations. Therefore, the number of GO terms that were significant in the gene set at the central part of tAg was small, but those terms could have various characteristics in each gene set.

Analysis of gene sets based on querying genes can reveal GO terms and pathways that contain many genes that show a similar or identical fluctuation pattern as the query gene. Many of the upregulated query genes were located in area D on the CA plot (Table 1, S6 Fig). Among those genes, *Cxcl5* was located close to straight line 1 and far from straight lines 2 and 3 on the

**Table 12. The number of GO terms belonging to each GO category and pathways that were significant in the GO and pathway analysis of the top 100 to 1000 gene sets whose distances were close to each query gene on the CA plot.**

| Query gene | Biological process | | | Cellular component | | | Molecular function | | | Pathway | | |
|---|---|---|---|---|---|---|---|---|---|---|---|---|
| | 100 set | 300 set | 1000 set | 100 set | 300 set | 1000 set | 100 set | 300 set | 1000 set | 100 set | 300 set | 1000 set |
| Cxcl5 | 468 | 391 | 1374 | 0 | 5 | 14 | 23 | 27 | 100 | 18 | 15 | 30 |
| Gm5483 | 468 | 391 | 1370 | 0 | 5 | 14 | 23 | 27 | 100 | 18 | 15 | 30 |
| Ccl4 | 468 | 390 | 1368 | 0 | 5 | 14 | 23 | 27 | 100 | 18 | 15 | 30 |
| Il-1β | 468 | 391 | 1370 | 0 | 5 | 14 | 23 | 27 | 100 | 18 | 15 | 30 |
| S100a8 | 247 | 505 | 1266 | 0 | 4 | 13 | 4 | 27 | 92 | 1 | 5 | 17 |
| Clec4d | 47 | 310 | 1351 | 0 | 2 | 14 | 7 | 20 | 100 | 1 | 5 | 29 |
| Slpi | 0 | 17 | 126 | 0 | 23 | 72 | 0 | 4 | 20 | 0 | 0 | 2 |
| Saa3 | 3 | 155 | 169 | 0 | 41 | 63 | 0 | 0 | 28 | 0 | 0 | 3 |
| Cd300lf | 10 | 379 | 1343 | 0 | 7 | 13 | 3 | 12 | 104 | 0 | 3 | 29 |
| Hs3st5 | 0 | 21 | 52 | 15 | 30 | 43 | 0 | 5 | 7 | 0 | 1 | 1 |
| Ddit4l | 0 | 0 | 25 | 0 | 6 | 39 | 1 | 0 | 8 | 0 | 0 | 1 |
| Efnb3 | 0 | 0 | 42 | 0 | 13 | 53 | 0 | 1 | 14 | 0 | 0 | 2 |
| Lzts2 | 0 | 0 | 34 | 0 | 0 | 35 | 1 | 2 | 6 | 0 | 0 | 1 |
| Slc26a10 | 0 | 9 | 65 | 0 | 7 | 50 | 0 | 3 | 9 | 0 | 0 | 2 |
| Fam83d | 0 | 0 | 12 | 0 | 0 | 28 | 0 | 3 | 8 | 0 | 0 | 0 |
| Ddit4l | 0 | 0 | 25 | 0 | 6 | 39 | 1 | 0 | 8 | 0 | 0 | 1 |
| Myh7 | 0 | 0 | 10 | 0 | 0 | 2 | 0 | 0 | 7 | 0 | 0 | 1 |
| Tet1 | 0 | 0 | 5 | 0 | 0 | 24 | 0 | 0 | 6 | 0 | 0 | 1 |
| Plcd4 | 8 | 20 | 65 | 19 | 33 | 54 | 0 | 4 | 16 | 1 | 6 | 12 |
| Ostn | 0 | 0 | 76 | 0 | 0 | 31 | 1 | 0 | 11 | 0 | 0 | 15 |
| Mettl11b | 0 | 31 | 58 | 1 | 27 | 65 | 0 | 3 | 11 | 0 | 6 | 10 |
| Gm6288 | 0 | 0 | 5 | 0 | 0 | 0 | 0 | 0 | 7 | 0 | 0 | 1 |
| Tmem233 | 13 | 40 | 78 | 25 | 43 | 65 | 4 | 4 | 11 | 6 | 7 | 9 |
| Arg1 | 34 | 95 | 322 | 0 | 0 | 38 | 7 | 2 | 37 | 0 | 0 | 11 |
| Ly6f | 0 | 0 | 0 | 0 | 0 | 0 | 1 | 0 | 0 | 0 | 0 | 0 |

CA plot (S6 and S7 Figs). Because each line divided the magnitude of the relationship of the FC values, we estimated that the values of FC6/FC24 and FC12/FC24 of *Cxcl5* were large, whereas that of FC6/FC12 was small (In fact, their values were 13.9, 9.9, and 1.4, respectively, calculated from Table 1). Therefore, *Cxcl5* and the genes located around it on the CA plot can be gene markers that were upregulated significantly in the early phase after injury.

FC6 of *Gm5483*, *Ccl4*, and *Il-1β* was smaller than that of *Cxcl5*, but their FC6/FC24 and FC12/FC24 were larger than those of *Cxcl5*, reflecting their farther location from straight lines 2 and 3 than *Cxcl5* (S7 Fig). *Ccl4* is involved in myoblast proliferation after skeletal muscle injury [45]. *Il-1β* is a cytokine produced by macrophages and myogenic cells [46, 47], and is upregulated after tissue injury [32–34]. *Gm5483*, a synonym for *Cstdc4*, had a small number of GO terms such as "cysteine-type endopeptidase inhibitor activity" [14], but its function in wound healing has not been studied in detail. Considering that its FC value was maintained more than 100 even at 24 h after injury, *Gm5483* was seemed to play a certain role in both the inflammatory response of early phase and the muscle repair of late phase after injury. Because of their high FC6 to FC24 ratio, these genes may be useful for distinguishing different times after injury. On the other hand, *S100A8* and *Cd300lf* were located closer to straight lines 2 and 3 than *Cxcl5* on the CA plot (S8 Fig), and the FC6/FC24 or FC12/FC24 of these genes was small compared to *Cxcl5*. *S100A8* is expressed in differentiating suprabasal wound keratinocytes [48], and *S100A8* mRNA levels are upregulated in skeletal muscle tissue during both Il-6

infusion and exercise [49]. Also, *Cd300lf* induces cell death and promotes phagocytosis [30, 31]. These genes may be early, upregulated markers in wounds, although they are unsuitable for distinguishing wounds created within 24 h.

*Clec4d*, which is expressed in monocytes and can induce phagocytosis and proinflammatory cytokine production [28], was located in area C, but close to *S100a8* and *Cd300lf* in area D on the CA plot (S8 Fig), and the values of FC6/FC24 and FC12/FC24 were also small. Therefore, *Clec4d* may be a gene marker, similar to *S100a8* and *Cd300lf*. *Slpi* located in area B is expressed by macrophages and neutrophils [36]. The absence of *Slpi* leads to delayed wound healing, an increased and prolonged inflammatory response, enhanced elastase activity, and delayed matrix accumulation [37]. *Saa3* is mainly secreted from the liver and macrophages, stimulates toll-like receptor 4 activity, and induces NF-κB activation, a hallmark of inflammation [41–43]. On the CA plot, these genes were located near point P compared to the other upregulated query genes (S9 Fig), and each FC value had a relatively small difference. *Slpi* and *Saa3* may be late markers that were upregulated in the wounds. However, these genes may be more useful as negative markers at the time point when expression returns to baseline. Further microarray investigations after 24 h are needed.

In addition, *Cd72*, which was not designated as a query gene in this study, was located in area A at coordinates (−0.014, 0.558) on the CA plot and is a negative regulator of B-cell responsiveness (S9 Fig) [50]. *Cd72* was located near the intersection of the line connecting *Arg1-Ly6f* and line 1 and far from straight lines 2 and 3, similar to *Cxcl5*. However, the fold-values of FC6, FC12, and FC24 were 0.485, 0.611, and 5.266, respectively, which were opposite those of *Cxcl5*. Therefore, *Cd72* and the genes located around it on the CA plot may be gene markers that were upregulated significantly in late wounds. Furthermore, *Cd72* was downregulated at 6 and 12 h, unlike *Cxcl5*, which was upregulated at all time points, suggesting that *Cd72* can be used as a marker that is downregulated early after injury. *Cd72*, a negative regulator of B-cell responsiveness, would have been downregulated until 12 h to promote the inflammatory response, and then turned to upregulation at 24 h to inhibit the excessive inflammatory response. Thus, the CA plot helps identify gene marker candidates that are useful for estimating the timing of injuries.

Most genes that showed a more than 5-fold upregulation in expression had GO terms associated with inflammatory reactions, cytokines, and wound healing. Among them, the GO terms associated with skeletal muscle and myoblasts were mainly significant at 6–12 h of more than 3-fold upregulated sets, which would be an early phase reaction to skeletal muscle injury. On the other hand, some GO terms such as muscle cell differentiation (GO: 0042692) were significant in the downregulated sets. This GO term was not significant in the more than 3-fold upregulated set at 12 h, which contained 47 genes belonging in the term, whereas it was significant in the more than 3-fold downregulated set containing 35 genes (Table 5). This would be due to the difference in the total number of genes in the sets; therefore, it was somewhat difficult to determine whether upregulation or downregulation was dominant. Nonetheless, at least for muscle-related GO terms, the downregulated genes appeared to increase in number at 24 h. In the "CC" category, GO terms associated with the cell membrane were significant, which should reflect, for example, cytokine receptor activation of the cell membrane by inflammation. Also, the GO terms and pathways that were significant in common with all upregulated sets suggests that many genes associated with inflammatory reactions and cell membranes maintained high expression (more than 5-fold) up to 24 h after injury. Only a small number of significant GO terms were present in the downregulated sets; several reasons for this are possible (Table 4). Some sets contained many genes with few or no GO term annotations in the set, or had various and divergent GO term annotations. There were also fewer significant pathways in the downregulated sets, presumably because for similar reasons as for

the GO terms (Table 4). In the gene set that showed more than 3-fold downregulation at 24 h, pathways related to myocardial disease and the endocrine system were significant. Many genes related to or characteristic of skeletal muscle injury were also included in the myocardial disease pathway. However, this pathway was only significant at 24 h and is thus less important for the investigation of gene expression after skeletal muscle injury.

Although some GO terms and pathways were significant at a specific time point, when considering the fold changes of gene sets, we consider that analysis with these sets will be insufficient for studying the time-dependent dynamics of gene expression. In GO and pathway analyses of gene sets that changed more than 3- and 5-fold, each analysis was performed on FC values at a single time point. However, some genes were not statistically significant at that time point even though the FC value was up- or down-regulated more than 5-fold, and these were not included in the sets. Also, the large differences in the number of genes belonging to each gene set may make comparisons of each set difficult.

In gene set analysis based on query genes, all genes that pass QC are included in the set, thus there is no possibility that important genes would be excluded by statistical processing. In addition, since the number of genes in each set is all adjusted, it may be convenient to compare GO terms and pathways that were significant among each set. Moreover, GO term and pathway fluctuations throughout the time period can be estimated more easily than by using FC values as thresholds. In the CA plot of this study, genes with FC values far from 1 tended to gather around each vertex or point P of tAg, so it was easy to identify GO terms and pathways that were greatly upregulated or downregulated at each time point. However, for example, not only genes with $FC24 \geq 1$, but also genes with $FC24 < 1$ are included in the Cxcl5 set, thus it is necessary to confirm whether significant GO terms or pathway activity were upregulated or downregulated at 24 h compared to the control. Analyses of gene sets based on CA plots will be useful for investigating time-dependent fluctuations after injury and may compensate for some problems in the gene set based on fold changes.

In order to elucidate factors that could contribute to the formation of the gene sets other than the gene score, we investigated the characteristics of significant GO terms and pathways in the gene sets related to the time series scores. On the biplot, the 3 time series scores located in the area where genes with the lowest FC value at a particular time point (for example, the coordinate score of 6 h was located in area b in Fig 6), and genes with large FC values were sparse in the neighborhood of the time series scores (Fig 2a–2c). As a result, the genes contained in each set should have been unchanged in expression throughout time or downregulated only at particular time points, and there were only a small number of significant GO terms in the gene sets. We estimated that genes with smaller FC values would tend to cluster around the particular time series score (Fig 2d–2f), and that significant GO terms would show a strong decrease in activity at a specific time.

FC values were converted with the arctangent function and indicated as degrees. Therefore, the value of 1 was represented as 45 degrees, and the converted value shifted from 0 to 90 as the FC value became smaller. Because CA was performed on these values of degrees, genes with a small FC or a high degree value at each time point were dominant in the vicinity of the particular time series score (Fig 2d–2f). In particular, genes with an FC value less than 0.2 at each time point were distributed radially from point P toward each time series score, and those at all time points were localized around point P (S5 Fig). Each time series score seemed to be located a little away from point P in the vicinity of the line that bisected the area where the genes with the highest degrees at the same time point gathered. In contrast, genes with a small degree or a large FC value at each time point were distributed throughout the area of tAg (Fig 2a–2c). Further examination is needed to clarify the positional relationship between them, and the time series scores need further examination.

## Conclusion

Visualization with CA, which can cluster genes with similar expression dynamics, is an informative method of time course analysis of cytokines. GO and pathway analysis can be applied on selected genes from the plot based on a few marker genes. In this study, GO terms and pathways related to inflammation, muscle, and so on were significant in some query gene sets, so it is useful to analyze microarrays by referring to gene expression dynamics on CA plots. One limitation of this experimental model, which follows the principle of reduction, is that the effects of skin injury could be only assessed at 0 h or in the control. Thus, the post-injury results should include interactions with skin wounds to a certain extent.

## Supporting information

**S1 Fig. Pathway map mmu04060 (Cytokine-cytokine receptor interaction) that was significant in some gene sets.** This pathway was also significant in the top 1,000 gene set whose distances are close to *Cxcl5* on the CA plot. In the *Cxcl5* gene set, the FC values of many genes belonging to this pathway are more than 1 throughout time course (shown in red filled frame), but FC24 of some genes is less than 1 (red frame). This is because *Cxcl5* was located near the vertex of tAg where genes with large FC values tended to gather, and this set consisted of genes distributed in the area of C and D in Fig 6 (FC24 was the smallest at all times). The figure was created with KEGG Mapper (https://www.genome.jp/kegg/mapper.html). 10.6084/m9. figshare.10327334.
(TIF)

**S2 Fig. Distribution of genes with magnitude relationship patterns of FC values at all time points on the CA plot.** Genes corresponding to the following conditions are shown in gray on the CA plot. a): FC6, FC12, FC24 < 1; b): FC12, FC24 < 1 < FC6; c): FC6, FC24 < 1 < FC12; d): FC6, FC12 < 1 < FC24; e): FC24 < 1 < FC6, FC12; f): FC6 < 1 < FC12, FC24; g): FC12 < 1 < FC6, FC24; h): 1 < FC6, FC12, FC24. Genes formed fairly clear clusters on the plot according to whether each of the 3 FC values was greater than 1. Genes that did not satisfy each condition and the 3 time series scores are shown in orange and black dots, respectively.
(TIF)

**S3 Fig. Distribution of genes that showed specific time-dependent expression patterns on the CA plot (see Table 2) other than the patterns shown in Fig 4a).** Plot of genes that have two time series significantly upregulated and one time series significant downregulated. Gray dots: (U, U, D); blue dots: (U, D, U); red dots: (D, U, U). b): Plot of genes that have one time series significantly upregulated and two time series without fluctuation. Gray dots: (U, -, -); blue dots: (-, U, -); red dots: (-, -, U). c): Plot of genes that have one time series significantly upregulated and two time series significantly downregulated. Gray dots: (U, D, D); blue dots: (D, U, D); red dots: (D, D, U). d): Plot of genes that have one time series significantly downregulated and two time series without fluctuation. Gray dots: (-, -, D); blue dots: (-, D, -); red dots: (D, -, -). e): Plot of genes that have two time series significantly downregulated and one time series without fluctuation. Gray dots: (-, D, D); blue dots: (D, -, D); red dots: (D, D, -). Genes with the same fluctuation direction are plotted at symmetrical positions with each other across the center of tAg according to the combination of fluctuation behavior at each time point. Genes that did not satisfy any condition and the 3 time series scores are shown in orange and black dots, respectively.
(TIF)

**S4 Fig. Distribution of genes for which the ratio of FC values of two time points were close to 1 among each cluster of genes (See Fig 5; located in areas A to F).** For each gene, (highest FC value among the three time points)/(middle FC value) and (middle FC value)/(lowest FC value) were calculated, and the top 100 genes whose ratio was close to 1 were selected and plotted. Blue dots: FC6/FC12 and FC12/FC24 were close to 1 (area A in Fig 5). Gray dots: FC24/FC6 and FC24/FC12 (area B). Yellow dots: FC24/FC6 and FC12/FC6 (area C). Red dots: FC24/FC12 and FC12/FC6 (area D). Purple dots: FC24/FC12 and FC24/FC6 (area E). Green dots: FC12/FC6 and FC24/FC6 (area F). In each area, selected genes were distributed along the two border lines. Black dots indicate 3 column scores.
(TIF)

**S5 Fig. Distribution of genes whose FC value was less than 0.2 at each time point (a) and at all time points (b).** Yellow dot indicates the approximate intersection of the six areas in Fig 6 on the CA plot (point P). a): Plot of genes with FC values less than 0.2 at a specific time point. Gray dots: 6 h; blue dots: 12 h; red dots: 24 h. b): Magnified image around point P; the 46 genes with FC values less than 0.2 at all time points are shown in gray dots. There was a tendency for genes with small FC values to gather around point P. Genes that did not satisfy any condition and the 3 time series scores are shown in orange and black dots, respectively.
(TIF)

**S6 Fig. Distribution of upregulated query genes (white diamonds) on the CA plot in Fig 6.** Magnified images around each upregulated query gene are shown in S7, S8 and S9 Figs. Black dots indicate the 3 time series scores.
(TIF)

**S7 Fig. Distribution of *Cxcl5*, *Gm5483*, *Ccl4*, and *Il-1β* on the CA plot in S6 Fig.** Other genes in the area of C and D, and straight line 1 are also shown.
(TIF)

**S8 Fig. Distribution of *S100a8*, *Cd300lf*, and *Clec4d* on the CA plot in S6 Fig.** Other genes in the area of C and D, and straight line 1 are also shown.
(TIF)

**S9 Fig. Distribution of *Slpi*, *Saa3*, and *Cd72* (larger dots than the other genes) on the CA plot in S6 Fig.** *Cd72* is shown for comparison with the other upregulated query genes. The 2 time series scores (black dots), other genes, and three straight lines are also shown in this figure.
(TIF)

**S1 Table. The number of significant GO terms included in each GO term of the next hierarchy of each GO category in GO analysis of each set in Table 3.**
(XLSX)

**S1 Appendix.**
(DOCX)

**S2 Appendix.**
(DOCX)

## Author Contributions

**Conceptualization:** Masataka Takamiya.

**Formal analysis:** Tetsuya Horita, Mamiko Fukuta, Hideaki Kato.

**Investigation:** Tetsuya Horita, Mohammed Hassan Gaballah, Sanae Kanno.

**Writing – original draft:** Tetsuya Horita.

**Writing – review & editing:** Yasuhiro Aoki.

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
