## [Decision Letter · Decision Letter 0]

3 Jan 2020

PONE-D-19-32433

Time course analysis of large-scale gene expression in incised muscle using correspondence analysis

PLOS ONE

Dear Dr. Horita,

Thank you for submitting your manuscript to PLOS ONE. After careful consideration, we feel that it has merit but does not fully meet PLOS ONE’s publication criteria as it currently stands. Therefore, we invite you to submit a revised version of the manuscript that addresses the points raised during the review process.  

Specifically, you need to address how these data help to better understand muscle injury.  Although this systematic microarray and correspondence analyses revealed changes of gene expression profiles for  large number of genes, and pathways involved,  however,  no injury time markers have been identified and no hypothesis was formulated and tested based on the data from this study, thus significantly diminishing the significance of this study.

We would appreciate receiving your revised manuscript by  April 3, 2020. To enhance the reproducibility of your results, we recommend that if applicable you deposit your laboratory protocols in protocols.io, where a protocol can be assigned its own identifier (DOI) such that it can be cited independently in the future. For instructions see: http://journals.plos.org/plosone/s/submission-guidelines#loc-laboratory-protocols

We look forward to receiving your revised manuscript.

Kind regards,

Shao Jun Du, Ph.D.

Academic Editor

PLOS ONE

Journal Requirements:

2. In your Methods section, please provide methods of euthanasia for the animal research.

Reviewers' comments:

Reviewer's Responses to Questions

**Comments to the Author**

1. Is the manuscript technically sound, and do the data support the conclusions?

Reviewer #1: Yes

Reviewer #2: Partly

2. Has the statistical analysis been performed appropriately and rigorously? 

Reviewer #1: Yes

Reviewer #2: Yes

3. Have the authors made all data underlying the findings in their manuscript fully available?

Reviewer #1: Yes

Reviewer #2: Yes

4. Is the manuscript presented in an intelligible fashion and written in standard English?

Reviewer #1: Yes

Reviewer #2: Yes

5. Review Comments to the Author

Reviewer #1: This manuscript provides a correspondence analysis of the time course of changes of gene expression from 6 h to 24 hours in incision-injured skeletal muscle using microarrays. It would be nice to use a schematic diagram that illustrate a sequence of pathological processes. Two more suggestions, do the qRT-PCR validation of the genes and compare qRT-PCR and microarray data on the time-dependent expression. List any post-operative medications, such as analgesics.

Reviewer #2: n this study, the time course of gene expression in injured skeletal muscle were detected and displayed, and preliminary analysis such as classification and annotation was done.

But the entire article is a simple listing and display of big data, which is of little significance; it does not further analyze the mechanism of skeletal muscle post injury and lacks clear and meaningful conclusions. Through the reading of this paper, it is impossible to make any impression, and it is impossible to see the significance of the research. Therefore, it is recommended that the authors extract the more meaningful parts based on the preliminary display of high-throughput data, and then do some more in-depth verification or research.

1. How to select the GO categories whose fold change are more than three or five times in the T5. What are the selection criteria?

2. How much down-regulation of pathway regulation and what aspects are involved?

3. In the changed pathway regulation, is it important to up-regulation or is it more meaningful to down-regulation? Is it important to have a significant change all the time, or is it important to have a significant change at just one stage?

4. How does CA analysis compare to another common PCA analysis?

5. What are the sorting criteria displayed by T5 and T6? P value or biological process?

6. Is gene set divided by fold change?

6. PLOS authors have the option to publish the peer review history of their article (what does this mean?). If published, this will include your full peer review and any attached files.

Reviewer #1: No

Reviewer #2: No

---

## [Author Response · Author response to Decision Letter 0]

19 Feb 2020

We thank the reviewers for their advice and suggestions. We have taken all comments into consideration and have rewritten the manuscript thoroughly. Our response to each comment and the revisions made in the manuscript are described below.

Editor

Response

We apologize for not sending the manuscript in the specified style. The manuscript has been corrected according to the PLOS ONE instructions.

2. In your Methods section, please provide methods of euthanasia for the animal research.

The following sentences were inserted into the corresponding part of the Methods.

“Materials and Methods (Animal treatment for DNA microarray)”, p5, L97-100, 

After surgery, the animals were allowed free access to food and water. At 6, 12, and 24 h after injury, mice were euthanized with a high concentration of carbon dioxide gas, and then a 3-mm thick sample of injured muscle tissue (about 30 mg) with the injury in the center was excised.

Response

A "Figure legends" section was created, and a caption for each figure is included there.

(from p50, L793 in the manuscript)

4.

“Specifically, you need to address how these data help to better understand muscle injury. Although this systematic microarray and correspondence analyses revealed changes of gene expression profiles for large number of genes, and pathways involved, however, no injury time markers have been identified and no hypothesis was formulated and tested based on the data from this study, thus significantly diminishing the significance of this study.”

To investigate whether gene markers for estimation of the timing of injuries can be queried more directly or visually from CA plots, not only from GO and pathway analysis, we have added the following descriptions to the Discussion, focusing on the features of the query genes on the CA plot. 

“Discussion”, p32-34, L503-553,

Analysis of gene sets based on querying genes can reveal GO terms and pathways that contain many genes that show a similar or identical fluctuation pattern as the query gene. Many of the upregulated query genes were located in area D on the CA plot (Table 1, Fig. S6). Among those genes, Cxcl5 was located close to straight line 1 and far from straight lines 2 and 3 on the CA plot (Fig. S6, S7). Because each line divided the magnitude of the relationship of the FC values, we estimated that the values of FC6/FC24 and FC12/FC24 of Cxcl5 were large, whereas that of FC6/FC12 was small (In fact, their values were 13.9, 9.9, and 1.4, respectively, calculated from Table 1). Therefore, Cxcl5 and the genes located around it on the CA plot can be gene markers that were upregulated significantly in the early phase after injury. 

FC6 of Gm5483, Ccl4, and Il-1β was smaller than that of Cxcl5, but their FC6/FC24 and FC12/FC24 were larger than those of Cxcl5, reflecting their farther location from straight lines 2 and 3 than Cxcl5 (Fig. S7). Ccl4 is involved in myoblast proliferation after skeletal muscle injury [44]. Il-1β is a cytokine produced by macrophages and myogenic cells [45,46], and is upregulated after tissue injury [31-33]. Gm5483, a synonym for Cstdc4, had a small number of GO terms such as “cysteine-type endopeptidase inhibitor activity” [14], but its function in wound healing has not been studied in detail. Because of their high FC6 to FC24 ratio, these genes may be useful for distinguishing different times after injury. On the other hand, S100A8 and Cd300lf were located closer to straight lines 2 and 3 than Cxcl5 on the CA plot (Fig. S8), and the FC6/FC24 or FC12/FC24 of these genes was small compared to Cxcl5. S100A8 is expressed in differentiating suprabasal wound keratinocytes [47], and S100A8 mRNA levels are upregulated in skeletal muscle tissue during both Il-6 infusion and exercise [48]. Also, Cd300lf induces cell death and promotes phagocytosis [29-30]. These genes may be early, upregulated markers in wounds, although they are unsuitable for distinguishing wounds created within 24 h.

Clec4d, which is expressed in monocytes and can induce phagocytosis and proinflammatory cytokine production [27], was located in area C, but close to S100a8 and Cd300lf in area D on the CA plot (Fig. S8), and the values of FC6/FC24 and FC12/FC24 were also small. Therefore, Clec4d may be a gene marker, similar to S100a8 and Cd300lf. Slpi located in area B is expressed by macrophages and neutrophils [35]. The absence of Slpi leads to delayed wound healing, an increased and prolonged inflammatory response, enhanced elastase activity, and delayed matrix accumulation [36]. Saa3 is mainly secreted from the liver and macrophages, stimulates toll-like receptor 4 activity, and induces NF-κB activation, a hallmark of inflammation [40-42]. On the CA plot, these genes were located near point P compared to the other upregulated query genes (Fig. S9), and each FC value had a relatively small difference. Slpi and Saa3 may be late markers that were upregulated in the wounds. However, these genes may be more useful as negative markers at the time point when expression returns to baseline. Further microarray investigations after 24 h are needed.

In addition, Cd72, which was not designated as a query gene in this study, was located in area A at coordinates (−0.014, 0.558) on the CA plot and is a negative regulator of B-cell responsiveness (Fig. S9) [49]. Cd72 was located near the intersection of the line connecting Arg1-Ly6f and line 1 and far from straight lines 2 and 3, similar to Cxcl5. However, the fold values of FC6, FC12, and FC24 were 0.485, 0.611, and 5.266, respectively, which were opposite those of Cxcl5. Therefore, Cd72 and the genes located around it on the CA plot may be gene markers that were upregulated significantly in late wounds. Furthermore, Cd72 was downregulated at 6 and 12 h, unlike Cxcl5, which was upregulated at all time points, suggesting that Cd72 can be used as a marker that is downregulated early after injury. Thus, the CA plot helps identify gene marker candidates that are useful for estimating the timing of injuries. 

Reviewer #1

1. Do the qRT-PCR validation of the genes and compare qRT-PCR and microarray data on the time-dependent expression.

Response

Thank you for your suggestion. qRT-PCR was performed for several genes in previous studies using RNA extracted from the injured muscle of mice at each time point [2,3]. The microarray results in this study were essentially consistent with those of qRT-PCR. For example, Cxcl5, Ccl4, and Il-1β were significantly upregulated compared to controls at the time selected. The following descriptions were added to the Discussion. 

“Discussion”, p29, L455-460,

In previous studies, the post-injury dynamics of 10 cytokines that were selected based on the data from a microarray analysis at 12 h post-injury were investigated with qRT-PCR from 6 to 24 or 48 h after injury using the same mouse incised wound model [2,3]. The results of the microarray analyses of the particular genes in this study were essentially consistent with those of qRT-PCR. Thus, we believe that the microarray for each time point in this study had fairly reliable reproducibility.

2. List any post-operative medications, such as analgesics.

Response

The animals were allowed free access to food and water without any specific treatment after the operation.

The following sentences were inserted into the corresponding part of the Methods.

“Materials and Methods (Animal treatment for DNA microarray)”, p5, L97-100, 

After surgery, the animals were allowed free access to food and water. At 6, 12, and 24 h after injury, mice were euthanized with a high concentration of carbon dioxide gas, and then a 3-mm thick sample of injured muscle tissue (about 30 mg) with the injury in the center was excised.

 

Reviewer #2

1. How to select the GO categories whose fold change are more than three or five times in the T5. What are the selection criteria?

Response

We considered that reactions such as inflammation and healing would begin following the muscle incisions, and thus, we selected GO terms related to these processes. We also focused on GO terms related to myoblasts during muscle repair. We did not use any other criteria for selecting GO terms.

The following sentence was added to the Results section:

“Results (Gene ontology analysis and pathway and gene set analysis)”, p12, L217-220, 

We mainly focused on the GO terms that were related to the processes of inflammation and wound healing involving myoblasts because we assumed that making the incision would initiate such processes.

2. How much down-regulation of pathway regulation and what aspects are involved?

3. In the changed pathway regulation, is it important to up-regulation or is it more meaningful to down-regulation? Is it important to have a significant change all the time, or is it important to have a significant change at just one stage?

Response

The answer to questions 2 and 3 is summarized below.

In the gene set that was downregulated more than 3-fold at 24 h, pathways related to myocardial disease and the endocrine system were significantly changed. Many genes related to or characteristic of skeletal muscle injury were also included in the myocardial disease pathway. However, this pathway was only significant at 24 hours and will be less important for the investigation of gene expression after skeletal muscle injury. 

Because of the statistical processing and　the large differences in the number of genes belonging to each gene set, we considered that analysis with gene sets based on fold changes would be insufficient for studying time-dependent dynamics of gene expression. In contrast, analyses of gene sets based on CA plots will be effective for investigation of the time-dependent fluctuation after injury and will compensate for problems in the gene set based on fold changes. 

Therefore, evaluation of not only the expression at a specific time point, but also all time fluctuations of the gene sets based on query genes in CA, will be important.

The following description was added to the corresponding part of the Discussion:

“Discussion”, p36-37, L575-603, 

In the gene set that showed more than 3-fold downregulation at 24 h, pathways related to myocardial disease and the endocrine system were significant. Many genes related to or characteristic of skeletal muscle injury were also included in the myocardial disease pathway. However, this pathway was only significant at 24 h and is thus less important for the investigation of gene expression after skeletal muscle injury. 

Although some GO terms and pathways were significant at a specific time point, when considering the fold changes of gene sets, we consider that analysis with these sets will be insufficient for studying the time-dependent dynamics of gene expression. In GO and pathway analyses of gene sets that changed more than 3- and 5-fold, each analysis was performed on FC values at a single time point. However, some genes were not statistically significant at that time point even though the FC value was up- or down-regulated more than 5-fold, and these were not included in the sets. Also, the large differences in the number of genes belonging to each gene set may make comparisons of each set difficult.

In gene set analysis based on query genes, all genes that pass QC are included in the set, thus there is no possibility that important genes would be excluded by statistical processing. In addition, since the number of genes in each set is all adjusted, it may be convenient to compare GO terms and pathways that were significant among each set. Moreover, GO term and pathway fluctuations throughout the time period can be estimated more easily than by using FC values as thresholds. In the CA plot of this study, genes with FC values far from 1 tended to gather around each vertex or point P of tAg, so it was easy to identify GO terms and pathways that were greatly upregulated or downregulated at each time point. However, for example, not only genes with FC24 ≥ 1, but also genes with FC24 < 1 are included in the Cxcl5 set, thus it is necessary to confirm whether significant GO terms or pathway activity were upregulated or downregulated at 24 h compared to the control. Analyses of gene sets based on CA plots will be useful for investigating time-dependent fluctuations after injury and may compensate for some problems in the gene set based on fold changes.

4. How does CA analysis compare to another common PCA analysis?

Response

According to Yano et al. [9], up- or down-regulated genes can be predicted only with CA using the arctangent function, and PCA is not appropriate for the clustering of genes according to their expression patterns using any index. Although the results of this microarray were not analyzed with PCA, CA using the arctangent function clearly clustered the genes according to the pattern of expression fluctuation over time.

Accordingly, we have added the following description to the corresponding part of the Discussion.

“Discussion”, p30, L473-476, 

They showed also that up- or down-regulated genes could be predicted only with CA using the arctangent function, and that PCA was not appropriate for the clustering of genes according to their expression patterns using any index [9].

5. What are the sorting criteria displayed by T5 and T6? P value or biological process?

Response

We did not use specific criteria for sorting. The tables related to GO or pathway analysis (Tables 5, 6, S1, S2A, C, E) were rearranged alphabetically by the GO term or name.

Accordingly, we have changed the corresponding titles as follows:

“Table 5”, p15-16, L248-251, “Table S1”, “S2 Table C”, p4-5, L33-35 (Appendix S2), 

Each table was arranged in ascending order according to the “GO term”. 

“Table 6”, p17, L253-254, “S2 Table A”, p1-2, L14-15 (Appendix S2), “S2 Table E”, p8-10, L58-59 (Appendix S2), 

Each table was arranged in ascending order according to “Name”.

6. Is gene set divided by fold change?

Response

As described in "Detection of upregulated or downregulated genes" in the manuscript, up- or down-regulated gene sets were created with genes that were significant with ANOVA and Tukey tests. Among the genes that were significant in these tests, genes with a FC more than 5 or less than 0.2 compared to the control at each time were extracted as more than 5-fold up- or down-regulated sets, respectively. Similarly, genes with a FC more than 3 or less than 0.33 compared to the control at each time were extracted as more than 3-fold up- or down-regulated sets, respectively. 

The sets that were up- or down-regulated more than 3-fold also included genes with an FC more than 5 and less than 0.2, respectively. This sentence was added to the manuscript.

Because a gene set based on the query gene is created by the distance from the query gene, FC is not involved in these gene sets.

Accordingly, we have added the following sentence to the corresponding part of the Methods as follows:

“Materials and Methods (Detection of upregulated or downregulated genes)”, p7, L150-152, 

Gene sets that were up- or down-regulated more than 3-fold also included genes whose FC was more than 5 and less than 0.2, respectively.

7.

 “But the entire article is a simple listing and display of big data, which is of little significance; it does not further analyze the mechanism of skeletal muscle post injury and lacks clear and meaningful conclusions. Through the reading of this paper, it is impossible to make any impression, and it is impossible to see the significance of the research. Therefore, it is recommended that the authors extract the more meaningful parts based on the preliminary display of high-throughput data, and then do some more in-depth verification or research.”

To investigate whether gene markers for estimation of the timing of injuries can be queried more directly or visually from CA plots, not only from GO and pathway analysis, we have added the following descriptions to the Discussion, focusing on the features of the query genes on the CA plot. 

“Discussion”, p32-34, L503-553,

Analysis of gene sets based on querying genes can reveal GO terms and pathways that contain many genes that show a similar or identical fluctuation pattern as the query gene. Many of the upregulated query genes were located in area D on the CA plot (Table 1, Fig. S6). Among those genes, Cxcl5 was located close to straight line 1 and far from straight lines 2 and 3 on the CA plot (Fig. S6, S7). Because each line divided the magnitude of the relationship of the FC values, we estimated that the values of FC6/FC24 and FC12/FC24 of Cxcl5 were large, whereas that of FC6/FC12 was small (In fact, their values were 13.9, 9.9, and 1.4, respectively, calculated from Table 1). Therefore, Cxcl5 and the genes located around it on the CA plot can be gene markers that were upregulated significantly in the early phase after injury. 

FC6 of Gm5483, Ccl4, and Il-1β was smaller than that of Cxcl5, but their FC6/FC24 and FC12/FC24 were larger than those of Cxcl5, reflecting their farther location from straight lines 2 and 3 than Cxcl5 (Fig. S7). Ccl4 is involved in myoblast proliferation after skeletal muscle injury [44]. Il-1β is a cytokine produced by macrophages and myogenic cells [45,46], and is upregulated after tissue injury [31-33]. Gm5483, a synonym for Cstdc4, had a small number of GO terms such as “cysteine-type endopeptidase inhibitor activity” [14], but its function in wound healing has not been studied in detail. Because of their high FC6 to FC24 ratio, these genes may be useful for distinguishing different times after injury. On the other hand, S100A8 and Cd300lf were located closer to straight lines 2 and 3 than Cxcl5 on the CA plot (Fig. S8), and the FC6/FC24 or FC12/FC24 of these genes was small compared to Cxcl5. S100A8 is expressed in differentiating suprabasal wound keratinocytes [47], and S100A8 mRNA levels are upregulated in skeletal muscle tissue during both Il-6 infusion and exercise [48]. Also, Cd300lf induces cell death and promotes phagocytosis [29-30]. These genes may be early, upregulated markers in wounds, although they are unsuitable for distinguishing wounds created within 24 h.

Clec4d, which is expressed in monocytes and can induce phagocytosis and proinflammatory cytokine production [27], was located in area C, but close to S100a8 and Cd300lf in area D on the CA plot (Fig. S8), and the values of FC6/FC24 and FC12/FC24 were also small. Therefore, Clec4d may be a gene marker, similar to S100a8 and Cd300lf. Slpi located in area B is expressed by macrophages and neutrophils [35]. The absence of Slpi leads to delayed wound healing, an increased and prolonged inflammatory response, enhanced elastase activity, and delayed matrix accumulation [36]. Saa3 is mainly secreted from the liver and macrophages, stimulates toll-like receptor 4 activity, and induces NF-κB activation, a hallmark of inflammation [40-42]. On the CA plot, these genes were located near point P compared to the other upregulated query genes (Fig. S9), and each FC value had a relatively small difference. Slpi and Saa3 may be late markers that were upregulated in the wounds. However, these genes may be more useful as negative markers at the time point when expression returns to baseline. Further microarray investigations after 24 h are needed.

In addition, Cd72, which was not designated as a query gene in this study, was located in area A at coordinates (−0.014, 0.558) on the CA plot and is a negative regulator of B-cell responsiveness (Fig. S9) [49]. Cd72 was located near the intersection of the line connecting Arg1-Ly6f and line 1 and far from straight lines 2 and 3, similar to Cxcl5. However, the fold values of FC6, FC12, and FC24 were 0.485, 0.611, and 5.266, respectively, which were opposite those of Cxcl5. Therefore, Cd72 and the genes located around it on the CA plot may be gene markers that were upregulated significantly in late wounds. Furthermore, Cd72 was downregulated at 6 and 12 h, unlike Cxcl5, which was upregulated at all time points, suggesting that Cd72 can be used as a marker that is downregulated early after injury. Thus, the CA plot helps identify gene marker candidates that are useful for estimating the timing of injuries.

---

## [Decision Letter · Decision Letter 1]

2 Mar 2020

PONE-D-19-32433R1

Time course analysis of large-scale gene expression in incised muscle using correspondence analysis

PLOS ONE

Dear Dr. Horita,

Thank you for submitting your manuscript to PLOS ONE. After careful consideration, we feel that it has merit but does not fully meet PLOS ONE’s publication criteria as it currently stands. Therefore, we invite you to submit a revised version of the manuscript that addresses the points raised during the review process.

We would appreciate receiving your revised manuscript by March 29, 2020. To enhance the reproducibility of your results, we recommend that if applicable you deposit your laboratory protocols in protocols.io, where a protocol can be assigned its own identifier (DOI) such that it can be cited independently in the future. For instructions see: http://journals.plos.org/plosone/s/submission-guidelines#loc-laboratory-protocols

We look forward to receiving your revised manuscript.

Kind regards,

Shao Jun Du, Ph.D.

Academic Editor

PLOS ONE

Reviewers' comments:

Reviewer's Responses to Questions

**Comments to the Author**

1. If the authors have adequately addressed your comments raised in a previous round of review and you feel that this manuscript is now acceptable for publication, you may indicate that here to bypass the “Comments to the Author” section, enter your conflict of interest statement in the “Confidential to Editor” section, and submit your "Accept" recommendation.

Reviewer #1: All comments have been addressed

Reviewer #2: All comments have been addressed

2. Is the manuscript technically sound, and do the data support the conclusions?

Reviewer #1: Yes

Reviewer #2: (No Response)

3. Has the statistical analysis been performed appropriately and rigorously? 

Reviewer #1: Yes

Reviewer #2: (No Response)

4. Have the authors made all data underlying the findings in their manuscript fully available?

Reviewer #1: Yes

Reviewer #2: (No Response)

5. Is the manuscript presented in an intelligible fashion and written in standard English?

Reviewer #1: Yes

Reviewer #2: (No Response)

6. Review Comments to the Author

Reviewer #1: The discussion should be revise using the result obtained and should not only be based on literature.

The author should add the references in the bibliography:

1.shen et al 2019 Front Physiol 10 (1298)

Reviewer #2: (No Response)

7. PLOS authors have the option to publish the peer review history of their article (what does this mean?). If published, this will include your full peer review and any attached files.

Reviewer #1: No

Reviewer #2: No

---

## [Author Response · Author response to Decision Letter 1]

6 Mar 2020

We thank the reviewers for their advice and suggestions. We have taken all comments into consideration and have rewritten the manuscript thoroughly. Our response to each comment and the revisions made in the manuscript are described below.

Reviewer #1

1. The discussion should be revise using the result obtained and should not only be based on literature.

Response

Thank you for your suggestion. We compared some of the references in the manuscript with the results of this study and added some discussion. 

The following descriptions were added to the Discussion, respectively. 

“Discussion”, p30-31, L484-487,

The large FC values of these genes in our microarray results were considered to be consistent with their bioactivity after injury (except for Gm5483, which is less well studied than other cytokines).

“Discussion”, p33, L524-526,

Considering that its FC value was maintained more than 100 even at 24 h after injury, Gm5483 was seemed to play a certain role in both the inflammatory response of early phase and the muscle repair of late phase after injury.

“Discussion”, p35, L558-560,

Cd72, a negative regulator of B-cell responsiveness, would have been downregulated until 12 h to promote the inflammatory response, and then turned to upregulation at 24 h to inhibit the excessive inflammatory response.

2. The author should add the references in the bibliography: 1.shen et al 2019 Front Physiol 10 (1298)

The suggested literature was added to the References. 

“References”, p44, L706-708, 

[20] Shen Y, Zhang R, Xu L, Wan Q, Zhu J, Gu J, et al. Microarray Analysis of Gene Expression Provides New Insights Into Denervation-Induced Skeletal Muscle Atrophy. Front Physiol. 2019 Oct 11;10:1298. doi: 10.3389/fphys.2019.01298. eCollection 2019.

Also, we have added the following description to the Discussion. 

“Discussion”, p30, L468-470, 

Shen et al. [20] performed PCA to visualize time-dependent expression pattern of microarray data of tibialis anterior muscle after peripheral nerve injury in rats.

---

## [Editor Report · Decision Letter 2]

9 Mar 2020

Time course analysis of large-scale gene expression in incised muscle using correspondence analysis

PONE-D-19-32433R2

Dear Dr. Horita,

We are pleased to inform you that your manuscript has been judged scientifically suitable for publication and will be formally accepted for publication once it complies with all outstanding technical requirements.

With kind regards,

Shao Jun Du, Ph.D.

Academic Editor

PLOS ONE
---

## [Editor Report · Acceptance letter]

11 Mar 2020

PONE-D-19-32433R2 

Time course analysis of large-scale gene expression in incised muscle using correspondence analysis 

Dear Dr. Horita:

I am pleased to inform you that your manuscript has been deemed suitable for publication in PLOS ONE. Congratulations! Your manuscript is now with our production department. 

With kind regards,

on behalf of

Dr Shao Jun Du 

Academic Editor

PLOS ONE